# Effects of *Lacticaseibacillus casei* LK-1 Fermentation on Physicochemical Properties, Chemical Compositions, Antioxidant Activities and Volatile Profiles of Pineapple Juice

**DOI:** 10.3390/foods14203474

**Published:** 2025-10-12

**Authors:** Shaodan Peng, Lian Yang, Wei Zhou, Yuan Yuan, Liangkun Liao, Xiaobing Huang, Chenghui Zhang, Xiao Gong, Jihua Li

**Affiliations:** 1Agricultural Products Processing Research Institute, Chinese Academy of Tropical Agricultural Sciences/Key Laboratory of Tropical Crop Products Processing of Ministry of Agriculture and Rural Affairs, Zhanjiang 524001, China; peony@catas.cn (S.P.);; 2School of Food Science and Engineering, Hainan University, Haikou 570228, China

**Keywords:** pineapple juice, *Lacticaseibacillus casei*, fermentation, antioxidant activity, volatiles

## Abstract

Lactic acid bacteria (LAB) are widely utilized in the production of various fermented foods. *Lacticaseibacillus casei* (*Ls. casei*) fermentation has been demonstrated to enhance antioxidant activity, flavor, and nutritional value. This study aimed to evaluate the physicochemical properties, proximate chemical compositions, antioxidant activities, and volatile profiles of pineapple juice fermented by *Ls. casei* LK-1. Strain growth, physicochemical parameters, phenolics and flavonoids, carbohydrates, organic acids (by HPLC), free amino acids (FAAs), antioxidant activities (DPPH and FRAP methods), and volatile profiles (by GC-MS) of fermented pineapple juice were characterized. After 30 h of fermentation, the cell count reached 9.07 log CFU/mL. The fermentation process led to a significant decrease in pH, total soluble solids (TSS), and the *a** value (*p* < 0.05), with an increase in the *L** and *b** values (*p* < 0.05). Lactic acid and total umami amino acids were elevated, whereas sucrose and total sweet amino acid levels remained unchanged. The fermented juice exhibited enhanced DPPH scavenging activity and FRAP values. In terms of volatile profiles, esters and terpenes were the dominant volatiles in the fermented pineapple juice. The fermentation resulted in the production of 2-heptanone, 2-undecanone, and a significant reduction in 2-furaldehyde (*p* < 0.05). These findings demonstrate the feasibility of using *Ls. casei* LK-1 for fermentation to develop novel fermented pineapple juice with improved antioxidant properties and a modified volatile profile.

## 1. Introduction

Pineapple (*Ananas comosus* (L.) Merr.), a tropical fruit that originates from South America, is renowned for its pleasant aroma [1]. Pineapple juice extracted from the pulp is a nutrient-rich beverage containing carbohydrates, vitamins, and minerals [2]. It also contains beneficial phenolic compounds, including chlorogenic acid, gallocatechin, and quercetin [3]. However, the color, aroma, and nutritional content of fresh pineapple juice can deteriorate over time [4]. Therefore, developing strategies to preserve or enhance the sensory qualities, nutritional value, and bioactive components of pineapple juice is important.

LAB are a group of Gram-positive bacteria known for producing lactic acid through carbohydrate fermentation and are widely utilized in the production of fermented foods [5]. *Ls. casei* is one of the most prominent bacteria among LAB and has been used for fermentation to enhance antioxidant activity, flavor, and nutritional value [6,7]. Previous research has found that lactobacilli fermentation increased amino acids and ash contents, reduced fat content, and improved pasting properties in yellow pea flours [8]. Similarly, fermentation with *Lactiplantibacillus plantarum* (*Lp. plantarum*) enhanced the antioxidant activity of apple juice and improved its flavor [9]. These findings indicate that LAB fermentation could be a viable strategy to minimize quality degradation in pineapple juice while potentially augmenting its bioactive properties.

*Ls. casei* LK-1, a potential probiotic isolated from traditional fermented pickles in our laboratory, has been shown to have good acid and bile salt tolerance, and maintained viability exceeding 8.5 log CFU/mL after 2 h of exposure to simulated gastrointestinal fluids [10]. Moreover, *Ls. casei* LK-1 exhibits exceptional adaptability and robust growth in challenging matrices. Previous studies have demonstrated that this strain possesses strong tolerance and excellent reproduction ability in high-sugar, high-acidity, and high-salt environments [11]. The strain has a short growth cycle, reaching the stationary phase within approximately 12 h of cultivation, and has demonstrated good acid productivity, yielding up to 16 g/L of lactic acid during the fermentation of pitaya juice [11]. These attributes collectively indicate good fermentation potential, rendering the strain particularly suitable for the fermentation of high-sugar and acidic fruit juices, such as pineapple juice.

Although LAB have been investigated for fruit juice fermentation, a comprehensive analysis of the effects of the *Ls. casei* strain on the multifaceted quality attributes of pineapple juice—particularly on the dynamic changes in key chemical components such as sugars, organic acids, and FAAs—is still limited. This investigation primarily aimed to analyze the effects of *Ls. casei* LK-1 fermentation on the physicochemical properties (pH, TSS, color), bioactive compounds (total phenolics, flavonoids), and detailed profiles of carbohydrates, organic acids, FAAs, and volatile compounds in pineapple juice. The results will provide detailed insights into the application of *Ls. casei* LK-1 in fruit juice fermentation.

## 2. Materials and Methods

### 2.1. Chemicals and Materials

Not-from-concentrate (NFC) pineapple juice (Jiaguoyuan, Goodfarmer Foods Holding (Group) Co., Ltd., Shanghai, China) was purchased from a local Walmart supermarket. According to the product label, the juice was 100% pure pineapple fruit juice. The nutritional composition per 100 mL was as follows: carbohydrates 12.3 g, protein 0 g, fat 0 g, and sodium 0 g. *Ls. casei* LK-1 was obtained from the Agricultural Product Processing Research Institute, Chinese Academy of Tropical Agricultural Sciences (Zhanjiang, Guangdong, China). Rutin (Ref. B20771), n-octadecane (Ref. B27381), gallic acid (Ref. B20851), malic acid (Ref. B20937), lactic acid (Ref. B67095), citric acid (Ref. B21313), and pyruvic acid (Ref. B24354) were from Yuanye Company (Shanghai, China). Acetic acid (Ref. A801303) was purchased from Macklin Company (Shanghai, China). Phosphoric acid (Ref. P433777) and DPPH (Ref. D141336) were purchased from Aladdin Company (Shanghai, China). Folin–Ciocalteu phenol reagent (Ref. F9252) and methanol (Ref. 34860-1L-R) were supplied by Sigma-Aldrich (St. Louis, MO, USA). The total antioxidant capacity (T-AOC) assay kit (Ref. BC1315), De Man-Rogosa-Sharp (MRS) agar (Ref. M8330), Plate Count Agar (Ref. P9270) and MRS broth (Ref. M8540) were obtained from Solarbio Company (Beijing, China). D-glucose/D-fructose (Ref. 12800) and sucrose/glucose/fructose (Ref. 12819) assay kits were purchased from BioSystems Company (Barcelona, Spain).

### 2.2. Preparation of Inoculum

*Ls. casei* LK-1 was activated in MRS broth under static conditions at 37 °C. The culture was then centrifuged at 1000× *g* for 5 min at room temperature and rinsed twice with sterile saline. Finally, it was resuspended in sterile distilled water to its original volume to prepare the bacterial inoculum.

### 2.3. Preparation of Fermented Pineapple Juice

Pineapple juice fermentation was conducted using the method of Lian et al. [11] and illustrated in Figure 1. The juice pH was adjusted from 3.8 to 6.8 using a 3 M NaOH solution, followed by pasteurization at 80 °C for 10 min. The control group (non-fermented pineapple juice) underwent the identical pretreatment. Prior to inoculation, aliquots of the sterilized pineapple juice were aseptically transferred to plate count agar plates and incubated at 37 °C for 48 h to verify the absence of microbial growth. After confirming sterility, 1% (*v*/*v*) of the *Ls. casei* LK-1 inoculum was added to the pineapple juice, which was then incubated continuously at 37 °C for 30 h. The control group was composed of pineapple juice cultured under the same conditions without the addition of fermentation starter. The entire procedure, from pretreatment to analysis, was carried out with three independent replicates.

### 2.4. Enumeration of Microorganisms

The viable cell counts were determined using the standard plate counting method on MRS agar [4]. Briefly, 1 mL of fermented or non-fermented juice was serially diluted in 9 mL of sterile saline to obtain serial dilutions ranging from 10^−1^ to 10^−6^. A 100 µL aliquot of an appropriate dilution was spread onto pre-poured MRS agar plates. After spreading, plates were inverted and incubated at 37 °C for 24 h. A negative control plate, spread with sterile saline only, was included in the assay. Only plates containing 30 to 300 colonies were used for counting, and the results were expressed as log colony-forming units (CFU) per milliliter (log CFU/mL). All experiments were performed in triplicate.

### 2.5. Determination of pH, TSS, and Color

The pH was measured with a pH meter (INESA PHS-3C, Shanghai, China). TSS was measured using a handheld refractometer (ATAGO PAL-3, Tokyo, Japan). Color was evaluated using a precision colorimeter (3NH NR20XE, Shenzhen, China), and results were represented as *L**, *a**, and *b** values [12]. All experiments were performed in triplicate.

### 2.6. Determination of Total Flavonoid Content (TFC) and Total Phenolic Content (TPC)

TPC was determined using the protocol outlined by Kwaw et al. [13]. Briefly, 0.6 mL of the sample was mixed with 3.5 mL of Folin–Ciocalteu reagent. After 10 min, 2.5 mL of a Na_2_CO_3_ (7.5 g/100 mL) solution was added. The mixture was then incubated in the dark for 1 h. Absorbance was measured at 760 nm using a spectrophotometer (Mapada M8, Shanghai, China). Gallic acid served as the reference standard, and distilled water served as the negative control. Results were expressed as mg of gallic acid equivalents (GAE) per liter of sample.

TFC was measured according to the method of Wu et al. [14]. 1 mL of the sample was added to 0.5 mL of NaNO_2_ (5 g/100 mL). After 6 min, 0.5 mL of Al (NO_3_)_3_ (10 g/100 mL) was added. After another 6 min, 4 mL of NaOH (1 M) and 6 mL of 40% ethanol were added. After 15 min, absorbance was measured at 510 nm using a spectrophotometer (Mapada M8, Shanghai, China). Rutin was used as the reference standard, and distilled water served as the negative control. Results were expressed as mg of rutin equivalents (RE) per liter of sample. All experiments were performed in triplicate.

### 2.7. Determination of Carbohydrates and Organic Acids

Total sugar, glucose, fructose, and sucrose concentrations were assessed using a Y15 Automatic Analyzer (BioSystems Company, Barcelona, Spain) in conjunction with D-glucose/D-fructose and sucrose/glucose/fructose assay kits. Briefly, samples were centrifuged at 5000× *g* for 5 min, and the supernatant was collected and diluted 25-fold with distilled water. The diluted supernatant was then analyzed following the manufacturer’s protocol. Measurements were performed at a detection wavelength of 340 nm and a reaction temperature of 37 °C, with distilled water serving as the blank control. All experiments were performed in triplicate.

Organic acids were determined using an HPLC system (Shimadzu, Kyoto, Japan) with a PR-C_18_ column (250 mm × 4.6 mm, 5.0 μm). The analytical procedure followed an improved method described by Xu et al. [15]. After centrifugation (5000× *g*, 10 min), the supernatant was diluted 5 times with distilled water and passed through a 0.22 μm polyethersulfone (PES) membrane filter (Tianjin Jinteng Experimental Equipment Co., Ltd., Tianjin, China). The column temperature was maintained at 30 °C. The mobile phase, consisting of 4% (*v*/*v*) phosphoric acid and methanol (95:5, *v*/*v*), was delivered isocratically at a flow rate of 0.5 mL/min. The detection was carried out at 210 nm with an injection volume of 10 μL. The runtime was 30 min. Quantification was achieved by comparison with external standards of known concentrations. All experiments were performed in triplicate.

### 2.8. Determination of FAAs

The sample was pretreated using the sulfosalicylic acid method [16]. Initially, 100 μL of 10% sulfosalicylic acid solution was added to 400 μL of the sample, which was then kept at 4 °C for 1 h. After incubation, the mixture was centrifuged at 8050× *g* for 15 min. The supernatant was collected and centrifuged again. The supernatant was diluted 1:5 (*v*/*v*) and filtered through a 0.22 μm PES filter (Tianjin Jinteng Experimental Equipment Co., Ltd., Tianjin, China). The FAAs were detected using an automatic amino acid analyzer (A300, membraPure GmbH, Berlin, Germany) equipped with a strong cation exchange (SCX) Column (4.0 mm × 125 mm, 5 µm) and a UV-Vis detector. The separation was achieved using lithium citrate buffer system as mobile phase, employing a multi-step linear gradient program at a flow rate of 0.3 mL/min. The column temperature was programmed as follows: initially set at 40 °C, increased to 70 °C over 15 min, and then held constant. The injection volume was 20 μL. The absorbance at 440 and 570 nm was monitored after postcolumn reaction with ninhydrin reagent at 130 °C. Each FAA was identified and quantified by comparing the peak area of each amino acid with that of the external standard. The internal standards included Cysteine (Cys), Gamma-aminobutyric acid (GABA), Arginine (Arg), Aspartic acid (Asp), Proline (Pro), Tyrosine (Tyr), Serine (Ser), Asparagine (Asn), Glutamine (Gln), Glycine (Gly), Alanine (Ala), Valine (Val), Methionine (Met), Isoleucine (Ile), Leucine (Leu), Lysine (Lys), Threonine (Thr), Phenylalanine (Phe). All experiments were performed in triplicate.

### 2.9. Determination of Antioxidant Activity

The FRAP was measured using the T-AOC assay kit(Beijing Solarbio Science & Technology Co., Ltd., Beijing, China). Briefly, 18 µL of FRAP working solution was mixed with 24 µL of sample, incubated at room temperature for 10 min, and the absorbance was measured at 593 nm using a multi-mode microplate reader (Synergy H1, BioTek Instruments, Inc., Winooski, VT, USA). A standard curve was prepared using a series of FeSO_4_·7H_2_O solutions. The DPPH scavenging activity was assessed following the protocol of Pontonio et al. [17] with minor adjustments. Briefly, 1 mL of a 0.1 mM DPPH–methanol solution was mixed with 1 mL of the diluted sample and reacted in the darkness for 30 min. Absorbance was measured at 517 nm, with the absorbance values of the blank (distilled water) and sample groups recorded as A_0_ and A_1_, respectively. The percentage of DPPH scavenging activity was calculated as follows: DPPH scavenging activity (%) = (A_0_ − A_1_)/A_0_ × 100

All experiments were performed in triplicate.

### 2.10. Volatiles Analysis

Volatile compounds were analyzed using GC-MS. A total of 3 mL of the sample was placed into a 15 mL vial. Then, 26 μL of octadecane (2 g/L) was added as the internal standard. The extraction of volatile compounds was conducted utilizing a solid phase microextraction (SPME) fiber (DVB/CAR/PDMS, 50/30 μm, Supelco, Bellefonte, PA, USA) at 45 °C for 15 min. An HP-5MS column (30 m × 0.25 mm, 0.25 μm, J&W Scientific, Folsom, CA, USA) was employed with the following temperature profile: starting at 45 °C for 2 min, increasing to 180 °C at 27 °C/min, then to 240 °C at 3 °C/min, and holding for 2 min. Helium was used as the carrier gas at 0.8 mL/min. Mass spectrometry was carried out with an ion source temperature of 230 °C, a mass range of 35–500 *m*/*z*, and 70 eV electron ionization. Volatile compounds were identified by comparing their mass spectra with those in the NIST14s/Wiley9 libraries, considering only matches with a similarity index greater than 85%. The relative concentration was calculated using the formula [18]: C_i_ = (C_is_ × A_i_)/A_is_ where C_i_ is the concentration of the analyte (μg/100 mL), C_is_ is the final concentration of the internal standard in the vial (μg/100 mL), A_i_ is the chromatographic peak area of the analyte, and A_is_ is the chromatographic peak area of the internal standard. Each treatment was repeated three times.

### 2.11. Statistical Analysis

Data were analyzed using Origin Pro 2024 SR1 (OriginLab Corporation, Northampton, MA, USA) and presented as mean ± standard deviation (SD). Data normality was confirmed using the Shapiro–Wilk test before further analysis. Statistical analysis was performed using one-way analysis of variance, and differences between sample means were evaluated with Tukey’s test at a significance level of *p* < 0.05. A polar heatmap with dendrograms was utilized to represent the volatile compounds, while principal component analysis (PCA) was employed to elucidate the variances and similarities in volatiles between fermented and non-fermented pineapple juice.

## 3. Results and Discussion

### 3.1. Changes in Viable Cell Counts During Fermentation

The viable cell count of *Ls. casei* LK-1 in pineapple juice increased rapidly within the first 12 h of fermentation, with no significant (*p* > 0.05) alterations observed thereafter (Figure 2A). During the 30 h fermentation period, the bacterial count increased notably from 6.72 ± 0.05 log CFU/mL to 9.07 ± 0.07 log CFU/mL (*p* < 0.05), exceeding the minimum recommended level for probiotics (10^6^ CFU/mL) [19]. The results indicated that *Ls. casei* LK-1 not only exhibited a robust growth rate in pineapple juice but also achieved a high final cell density, confirming its suitability for this fermentation medium. This finding is consistent with a previous report, where the viable cell count of *Ls. casei* in fermented pineapple juice reached up to 8.65 log CFU/mL under optimal conditions [20].

### 3.2. Changes in Physicochemical Properties During Fermentation

pH is a crucial factor because it can influence both product quality and microbial viability. During the first 12 h, the pH declined sharply, followed by a more gradual decline (Figure 2B). By the 30-h mark of fermentation, the pH had dropped to 3.81, which was remarkably lower than that of the control (*p* < 0.05). The starter culture utilized carbohydrates during its growth and metabolism to produce organic acids. The reduction in pH was primarily attributed to the accumulation of organic acids, particularly lactic acid. This trend was consistent with a previous study on a barley beverage fermented with *Ls. casei* [21].

The TSS content exhibited a gradual decline throughout the fermentation process, with no alterations in the control group (Figure 2C). The TSS decreased dramatically from an initial 12.56 to 11.89 Brix after fermentation (*p* < 0.05). TSS is often associated with soluble sugars in fruit juices. Therefore, the reduction in TSS content could primarily be attributed to the LAB using the sugars in the pineapple juice for cellular growth and converting them into lactic acid [22].

Color is a significant sensory attribute of the beverage, influencing consumer acceptance. As shown in Figure 2, noticeable color differences were observed between the fermented pineapple juice and the control group. Specifically, the *a** value of the fermented pineapple juice declined during fermentation, with the final *a** value measuring 1.44 ± 0.06, notably lower than the initial measurement at 0 h (*p* < 0.05, Figure 2D). Meanwhile, the *L** and *b** values increased rapidly during the first 12 h of fermentation, and then gradually stabilized (Figure 2E,F). These findings indicated that fermentation with *Ls. casei* LK-1 enhanced the brightness and yellowness of the pineapple juice while diminishing its redness. This color shift towards a brighter, more yellow appearance is often associated with enhanced visual quality and consumer appeal [23]. Color parameters are closely linked to pigment constituents such as anthocyanins and carotenoids. Previous research identified a strong positive correlation between the *a** value and cyanidin-3-O-(6″-O-malonyl-glucoside), cyanidin-3-O-(3″,6″-O-dimalonyl-β-glucopyranoside), and total anthocyanins, while the *b** value showed a strong positive correlation with lutein and total carotenoids [24,25]. A negative correlation has been observed between the *L** value and anthocyanins [25]. Thus, the decrease in the *a** value and increase in the *L** value in this study might result from the reduction in total anthocyanins. The changes in the content or structure of anthocyanins may be attributed to the decrease in pH and microbial activity during fermentation [12,26]. The increase in the *b** value might result from the release of lutein or carotenoid components influenced by physical or chemical factors [12].

### 3.3. Changes in Total Phenolics and Flavonoids During Fermentation

Phenolic compounds are important active ingredients in fruits and vegetables, known for their health benefits. A decline was observed in the TPC of both the fermented and control groups throughout the fermentation process (Figure 3A). TPC in the fermented juice decreased from 508.62 ± 1.49 to 472.29 ± 14.09 mg GAE/L, whereas that in the control group decreased to 467.90 ± 4.19 mg GAE/L. Additionally, TFC in the fermented juice decreased markedly after 12 h of fermentation (*p* < 0.05), with no significant alterations in the control group (*p* > 0.05, Figure 3B). Specifically, TFC in the fermented juice dropped from 78.21 ± 2.68 to 57.35 ± 5.92 mg RE/L. Wang et al. [27] reported similar findings, noting a gradual decrease in flavonoid content during the fermentation of pear juice with LAB. Previous studies have indicated that enzymes (e.g., glycosidases, polyphenol oxidases) produced by LAB could promote the degradation or transformation of phenolic compounds [12,28]. Therefore, it is hypothesized that the observed decrease in TPC and TFC could be attributed to the transformation of phenolic compounds by enzymes produced by the fermenting strain.

### 3.4. Changes in Carbohydrates During Fermentation

Carbohydrates are the primary carbon sources for microbial growth and are critical for assessing the quality of fermentation products. A decrease in the contents of total sugars, glucose, and fructose during the fermentation process was observed (Figure 4A–C), while the sucrose content remained stable (Figure 4D). The results suggested that *Ls. casei* LK-1 primarily utilized glucose and fructose during fermentation, with minimal utilization of sucrose. After fermentation, the total sugar concentration decreased from 116.92 ± 3.64 g/L to 93.92 ± 6.87 g/L (Figure 4A). This reduction in total sugar content was primarily due to the consumption of monosaccharides such as glucose and fructose. Compared to disaccharides and polysaccharides, *Ls. casei* demonstrated a preference for utilizing monosaccharides, particularly glucose [29], which might explain the lower utilization of sucrose by *Ls. casei* LK-1. Similar reductions in total sugar content were observed during the fermentation of kiwifruit juices by *Lacticaseibacillus* [30].

### 3.5. Changes in Organic Acids During Fermentation

Organic acids serve as secondary carbon sources for LAB during fermentation [15]. The level of lactic acid elevated during the fermentation of pineapple juice, reaching 22.05 ± 0.94 g/L after 30 h (Figure 5A). This indicated that *Ls. casei* LK-1 had excellent acid production abilities. A sharp increase in lactic acid levels was also observed during the fermentation of cloudy apple juice by LAB [31]. Lactic acid is the primary metabolite produced during LAB fermentation and can be synthesized through sugar fermentation, polyalcohol metabolism, and acid conversion [5]. It contributes to the tart flavor and functional properties of LAB-fermented products. The substantial production of lactic acid by *Ls. casei* LK-1 not only helped maintain the quality of the fermented juice but also enhanced its characteristic lactic flavor. It is important to note that this study quantified total lactic acid; future research should differentiate between L-lactic acid and D-lactic acid isomers to fully assess the product’s nutritional quality.

As fermentation progressed, the concentrations of malic and citric acids in the fermented pineapple juice gradually declined (Figure 5B,C), indicating that *Ls. casei* LK-1 could metabolize these acids. Liu et al. [32] reported similar reductions in malic and citric acid levels in fermented sea buckthorn juice using LAB. The reduction in malic and citric acid levels can be attributed to their conversion into lactic acid, acetate, or acetoin [5]. Pyruvic acid exhibited a sharp decline during the initial 12 h of fermentation, followed by a slight increase (Figure 5D). As a crucial intermediate in various metabolic pathways, the rate of pyruvic acid consumption might lag behind its production, leading to its accumulation after 12 h of fermentation. Furthermore, acetic acid was detected in the fermented pineapple juice, with its content notably decreasing after 12 h of fermentation (Figure 5E). Acetic acid is linked to a vinegar-like aroma and flavor, and an overpowering vinegary aroma could generate an unpleasant odor. It was important to note that, besides lactic acid, alterations in the levels of other organic acids were relatively slow or insignificant after 12 h of fermentation, possibly due to the inhibition of relevant metabolic enzyme activity induced by the low pH.

### 3.6. Changes in FAAs During Fermentation

FAAs were identified in the pineapple juices (Table 1). During the 30-h fermentation period, the total FAAs content in the fermented pineapple juice gradually decreased from 106.48 ± 3.57 mg/100 mL to 82.40 ± 2.59 mg/100 mL, while no significant alterations were observed in the control group. The enzymatic activities in the LAB fermentation environment accelerated amino acid metabolism, leading to dynamic changes in the total FAAs as fermentation progressed [4].

During the fermentation of pineapple juice, the distribution of FAAs was notably influenced by *Ls. casei* LK-1, as evidenced by changes in the 19 FAAs (Figure 6). Amino acids can be categorized into three taste groups: umami (Glu and Asp), sweet (Met, Ala, Gly, Pro, and Ser), and bitter (His, Lys, Val, Trp, Tyr, Phe, Ile, Arg, and Leu) [33]. As fermentation progressed, there was a noticeable increase in umami free amino acids, specifically Asp (Figure 6A). Sweet free amino acids like Ser, Ala, and Met decreased after fermentation (Figure 6B). However, a significant increase in Proline (Pro) levels resulted in no statistically significant net change (*p* > 0.05) in the total content of sweet free amino acids between 0 h and 30 h. Furthermore, *Ls. casei* LK-1 fermentation resulted in a reduction in most bitter free amino acids, such as Tyr, Val, Leu, and Lys (Figure 6C). Unflavored amino acids like Asn, Gln, and Thr also decreased during fermentation (Figure 6D). The decrease in amino acids could be attributed to LAB metabolism to aldehydes, carboxylic acids, and/or alcohols [34]. Conversely, the rise in amino acid levels might stem from the degradation of cellular proteins from dead *Ls. casei* LK-1 during fermentation or the interconversion of amino acids facilitated by *Ls. casei* enzymes. Overall, pineapple juice fermentation led to higher umami amino acid levels, stable sweet amino acid content, and a gradual decrease in bitter amino acids, suggesting that the taste of pineapple juice could be enhanced through fermentation with *Ls. casei* LK-1.

### 3.7. Changes in Antioxidant Activity During Fermentation

The FRAP value of the fermented pineapple juice exhibited little change before 24 h of fermentation, but increased significantly at 30 h (*p* < 0.05). In contrast, the control group showed a decreasing trend in FRAP value during fermentation (Figure 7A). These results suggested that while the fermentation process initially had a negative impact on FRAP—possibly due to temperature effects, fermentation with *Ls. casei* LK-1 significantly improved the FRAP of pineapple juice. By the end of fermentation, the FRAP of the fermented pineapple juice reached 115.54 ± 8.66 Fe^2+^ μmol/mL, signifying a 13.4% increase compared to the initial reading at 0 h. The DPPH scavenging activity of the fermented juice increased gradually as fermentation progressed, with significant differences (*p* < 0.05) observed from 24 h onwards compared to the 0 h time point, while the control showed no dramatic changes (Figure 7B). A study demonstrated that *Lp. plantarum* fermentation positively influenced the antioxidant activities of *Pleurotus eryngii* [35], which aligned with our findings. The antioxidant activities of foods are closely associated with phenolic compounds, some of which exist in conjugated forms with glycosides and sugars [36]. The increase in DPPH scavenging capability and FRAP might be attributed to the production of free aglycones with a stronger proton-supplying capacity or lower hydroxyl steric resistance during LAB fermentation [9,37]. Interestingly, the observed increase in antioxidant activities occurred alongside a decrease in total phenolics and flavonoids. This apparent discrepancy could be explained by the transformation of complex phenolic compounds into simpler, more bioaccessible phenolic acids or aglycones with higher antioxidant potency through microbial enzymatic activity (e.g., β-glucosidase) [38]. The release of other non-phenolic antioxidants, such as certain peptides or Maillard reaction products, might also contribute [39].

### 3.8. Volatile Profiles in Pineapple Juice After Fermentation

The effect of aroma compounds on the sensory characteristics of foods and consumer preferences is well-established. In this study, a total of 53 volatile components were identified in the fermented pineapple juice, including 22 esters, 15 terpenoids, 4 ketones, 1 aldehyde, 1 acid, 3 alcohols, and 7 other compounds (Table 2). Among them, ester volatile compounds are the primary contributors to the typical fruity and sweet aroma of pineapple juice. The content of most esters in fermented pineapple juice decreased significantly (*p* < 0.05) compared to the non-fermented pineapple juice. For instance, ethyl caproate, which is associated with pineapple- and apple-like aromas, decreased from 829.13 μg/100 mL to 549.96 μg/100 mL. Similarly, isoamyl acetate and ethyl butyrate, which contribute to sweet and banana-like scents, declined from 103.08 μg/100 mL to 85.60 μg/100 mL and from 99.26 μg/100 mL to 39.70 μg/100 mL, respectively. Corresponding reductions were also observed for other esters such as methyl caproate and ethyl caprylate, known for their fruity and wine-like aromas. The decrease in ester content may be attributed to ester hydrolysis caused by the high esterase activity produced by LAB during fermentation [40]. Similar phenomena have also been reported in melon juice and cashew juice fermented by *Ls. casei* [41]. The reductions in these key ester compounds suggest a potential attenuation of the typical intense fruity and sweet aroma profile after fermentation by *Ls. casei* LK-1, which could lead to an overall milder and softer flavor. Notably, a few esters, such as methyl 2-methylbutyrate and methyl 3-(methylthio) propionate, showed no significant difference (*p* > 0.05) in content between fermented and non-fermented pineapple juice, suggesting that these compounds may serve as stable contributors to the fruity and tropical pineapple-like aroma in the fermented juice.

Terpenoids are important sources of floral and woody aromas in fruits. Most terpenes, such as caryophyllene and valencene, showed no significant changes before and after fermentation (*p* > 0.05), indicating that *Ls. casei* LK-1 fermentation had minimal impact on them. These stable terpenoids help maintain the fundamental floral and fruity aroma characteristics of pineapple juice. In addition, some terpenoids, such as guaiene, α-muurolene, and delta-cadinene, were accumulated during the fermentation. These terpenes typically carry woody, spicy, and earthy notes, and their increase could potentially contribute to a more complex aroma profile in the fermented pineapple juice.

Methyl ketones like 2-heptanone, 2-nonanone, and 2-undecanone, were crucial volatiles produced during fermentation. They are produced mainly by the oxidation of unsaturated fatty acids under the action of LAB and may involve the Maillard reaction [42]. These compounds can contribute fruity, oily, and floral aromas, thereby potentially further enriching the overall flavor profile [43]. Conversely, the significant reduction of 2-furaldehyde (*p* < 0.05), a compound that contributes a roasted, almond-like, caramel-like aromatic profile but can impart an undesirable burnt odor at high concentrations, suggests a potential improvement in the flavor profile [44]. This reduction is likely due to the active enzyme system of LAB during fermentation, which converts aldehyde compounds into their corresponding carboxylic acids [45]. Hexanoic acid, known for its cheesy, rancid, and sweaty flavor [46], increased significantly after fermentation by *Ls. casei* LK-1 (*p* < 0.05). The impact of this increase on the overall sensory profile warrants further investigation, as it could influence consumer acceptability depending on its concentration relative to the perceptual threshold. Regarding alcohols, the presence of trans-geraniol with its pleasant floral character, and 1,2,3-butanediol may help enhance the aromatic complexity and potentially act as a solvent for other aromatic substances [27,47,48], thereby potentially improving the integration and perception of desirable flavor notes.

PCA of volatile profiles revealed a clear separation between the fermented and control pineapple juice samples (Figure 8B). The first two principal components (PC1: 64.0%, PC2: 16.9%) cumulatively accounted for 80.9% of the variance, strongly supporting the distinct separation between the fermented and control groups. Among the 53 volatile components, 2-undecanone, 2-nonanone, 2-heptanone, 1-octen-3-yl-acetate, heptanoic acid, ethyl ester, guaiene, and hexanoic acid exhibited strong correlations with fermented pineapple juice. These components were considered key variables post-fermentation.

## 4. Conclusions

In this work, *Ls. casei* LK-1 demonstrated robust growth and reproduction in pineapple juice fermentation. The fermentation process led to a decrease in pH, TSS, and *α** values, while increasing *L** and *b** values. Concurrently, a substantial increase in lactic acid concentration was observed. Various changes were observed, including reductions in the contents of flavonoids, glucose, fructose, citric acid, malic acid, pyruvic acid, acetic acid, and total bitter amino acids, while the concentrations of lactic acid and total umami free amino acids increased. Sucrose and total content of sweet amino acids showed no significant change throughout fermentation. The fermented pineapple juice exhibited heightened DPPH scavenging activity and FRAP. The primary volatile compounds were esters and terpenes, contributing to a fruity, floral, and woody flavor profile. 2-heptanone, 2-nonanone, and 2-undecanone were detected after fermentation, and 2-furaldehyde level was reduced. Overall, the fermentation process with *Ls. casei* LK-1 produced pineapple juice with modified physicochemical properties, enhanced DPPH scavenging activity and FRAP, and an altered volatile profile. This study provides a comprehensive foundation for the application of the potential probiotic strain *Ls. casei* LK-1 in fruit juice fermentation. However, this study has limitations, primarily the absence of turbidity analysis, sensory evaluation to confirm perceived quality improvements and shelf life testing to assess product stability. Future research should focus on conducting sensory analysis, evaluating in vivo bioactivity, and assessing the product’s shelf life to facilitate the development of consumer-acceptable functional fermented beverages.

## Figures and Tables

**Figure 1 foods-14-03474-f001:**
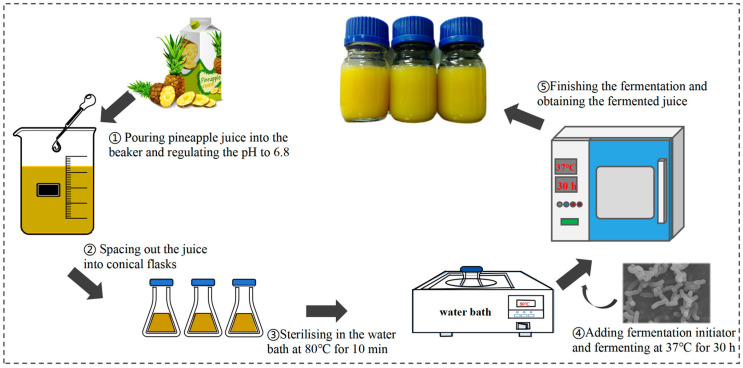
Schematic diagram of the fermentation process and sample preparation.

**Figure 2 foods-14-03474-f002:**
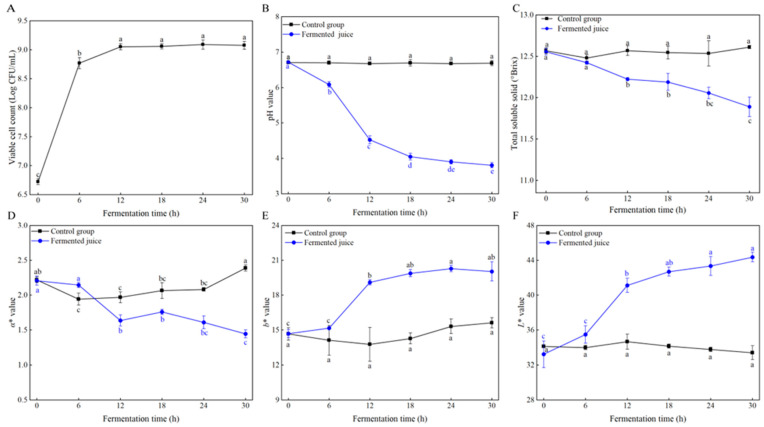
Changes in viable cell count (**A**), pH (**B**), TSS (**C**), and color parameters (**D**–**F**) during the fermentation. Values are mean ± SD (*n* = 3). Lowercase letters indicate significant differences over time (*p* < 0.05).

**Figure 3 foods-14-03474-f003:**
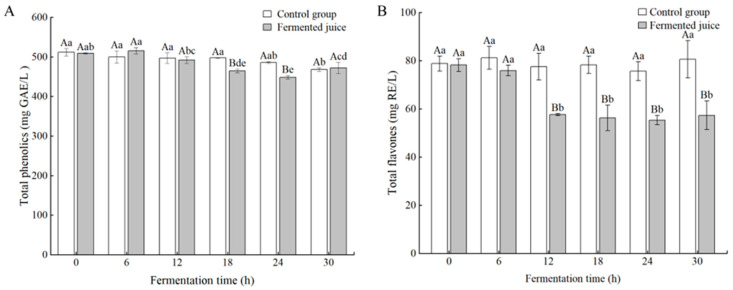
Changes in total phenolics (**A**) and total flavonoids (**B**) during the fermentation. Lowercase letters indicate significant differences over time (*p* < 0.05); capital letters indicate differences between samples at the same time point (*p* < 0.05).

**Figure 4 foods-14-03474-f004:**
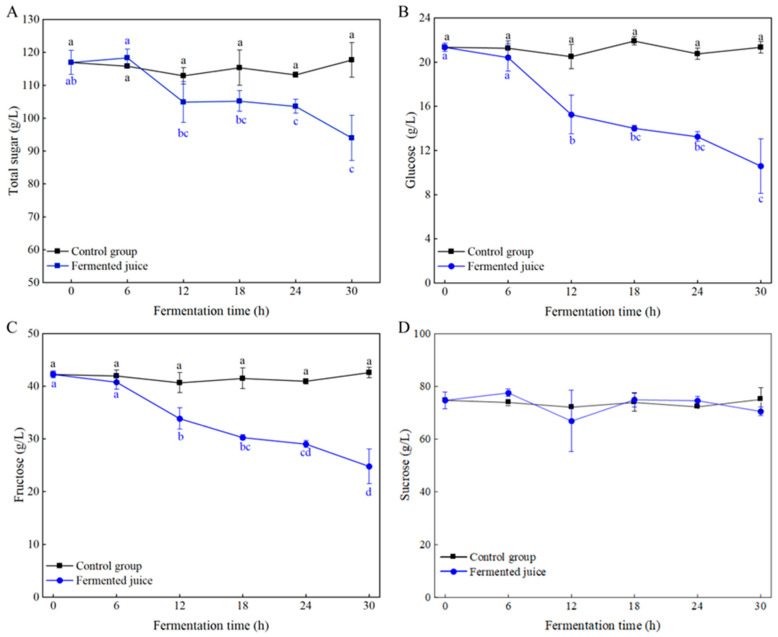
Changes in the content of total sugars (**A**), glucose (**B**), fructose (**C**), and sucrose (**D**) during the fermentation. Values are mean ± SD (*n* = 3). Lowercase letters indicate significant differences over time (*p* < 0.05).

**Figure 5 foods-14-03474-f005:**
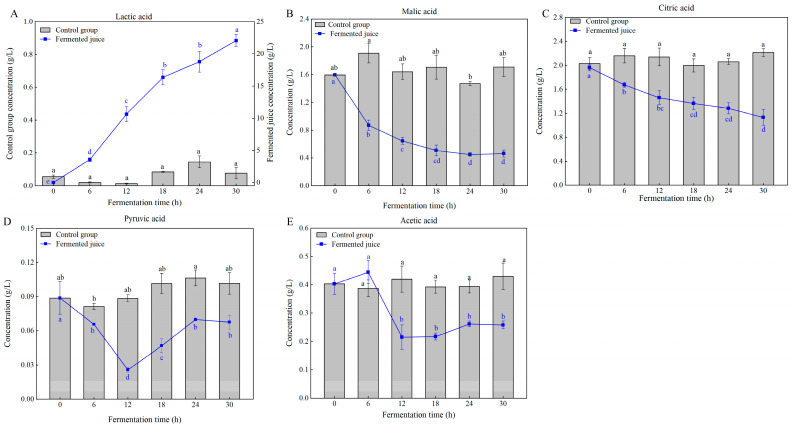
Changes in organic acids during the fermentation. Values are mean ± SD (*n* = 3). Lowercase letters indicate significant differences over time (*p* < 0.05). (**A**), lactic acid; (**B**), malic acid; (**C**), citric acid; (**D**), pyruvic acid; (**E**), acetic acid.

**Figure 6 foods-14-03474-f006:**
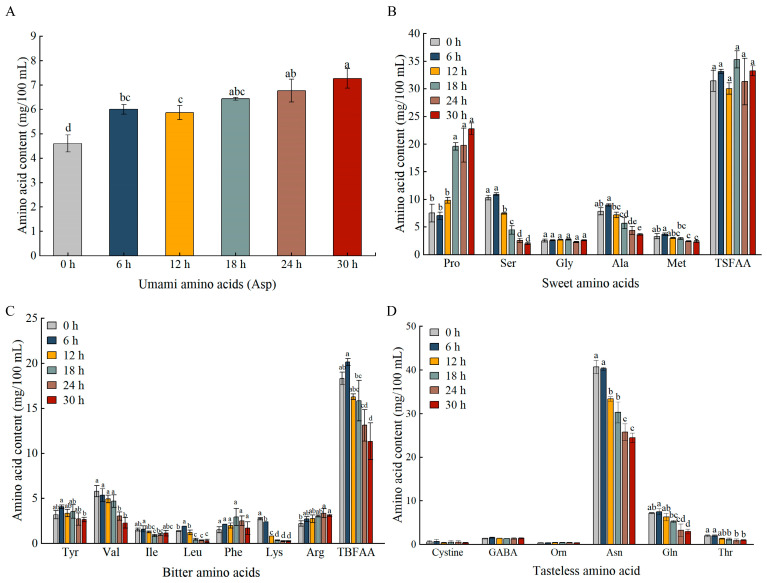
Changes in umami (**A**), sweet (**B**), bitter (**C**), and unflavored (**D**) amino acid content in the fermented pineapple juice. Values are mean ± SD (*n* = 3). Lowercase letters indicate significant differences over time (*p* < 0.05).

**Figure 7 foods-14-03474-f007:**
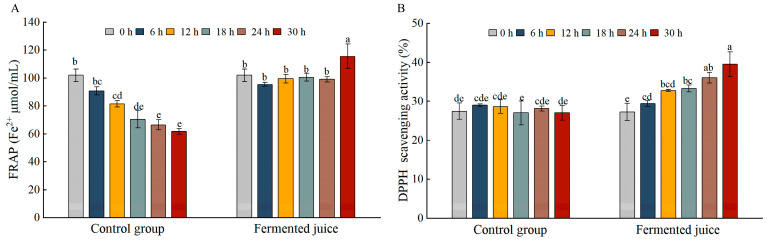
Changes in FRAP (**A**) and DPPH (**B**) scavenging activity during the fermentation. The error bars represent the standard deviations obtained from three independent experiments. Values are mean ± SD (*n* = 3). Lowercase letters indicate significant differences over time (*p* < 0.05).

**Figure 8 foods-14-03474-f008:**
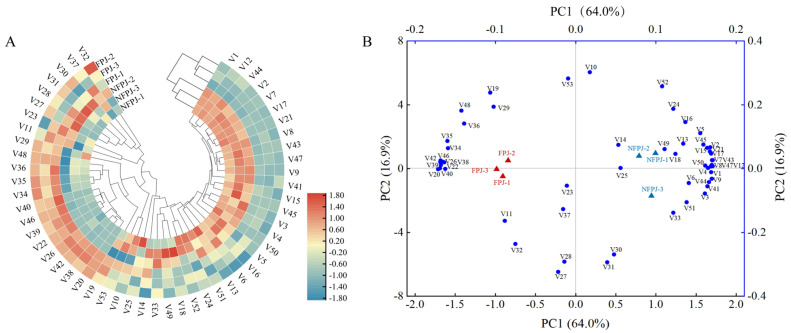
Volatile compound profiling to differentiate non-fermented and fermented pineapple juice: (**A**) cluster heatmap displaying correlation coefficient (r) represented by distinct color intensities; the legend on the right indicates the color scale corresponding to r values. (**B**) PCA: The black axis represents the score plot, while the blue axis corresponds to the loading plot. FPJ-1, FPJ-2, and FPJ-3 denote fermented pineapple juice (FPJ) samples numbered 1, 2, and 3, respectively, whereas NFPJ-1, NFPJ-2, and NFPJ-3 refer to the corresponding non-fermented pineapple juice (NFPJ) samples with the same numbering. Key volatile compounds (V1–V53) are listed. V1, Methyl 2-methylbutyrate, V2, Ethyl butyrate, V3, Ethyl 2-methylbutyrate, V4, Isoamyl acetate, V5, Ethyl valerate, V6, Methyl caproate, V7, Methyl (E)-hex-3-enoate, V8, Ethyl caproate, V9 Ethyl 3-hexenoate, V10, Methyl 3-(methylthio)propionate, V11, tert-Butyl propionate, V12, Methyl 2-methylacetoacetate, V13, Ethyl 3-methylthiopropionate, V14, 4-Octenoic acid methyl ester, V15, Methyl caprylate, V16, Ethyl (Z)-oct-4-enoate, V17, Ethyl caprylate, V18, Methyl 3-acetoxyhexanoate, V19, Methyl (Z)-4-decenoate, V20, 1-Octen-3-yl-acetate, V21, Decanoic acid, ethyl ester, V22, Heptanoic acid, ethyl ester, V23, beta-(Z)-Ocimene, V24, (E)-3,7-dimethylocta-1,3,6-triene, V25, Caryophyllene, V26, Guaiene, V27, Copaen, V28, beta-ylangene, V29, Germacrene D, V30, (+)-Sativene, V31, (−)-Clovene, V32, Valencene, V33 β-Selinene, V34, ɑ-Muurolene, V35, delta-Cadinene, V36, Guaiazulene, V37, (−)-Aristolene, V38, 2-Heptanone, V39, 2-Nonanone, V40, 2-Undecanone, V41, 2,5-Dimethyl-4-methoxy-3(2H)-furanone, V42, Hexanoic acid, V43, 2-Furaldehyde, V44, trans-Geraniol, V45, 1,6-Hexanediol, V46, 1, 2, 3-Butanetriol, V47, (3E,5E)-2,6-Dimethyl-1,3,5,7-octatetrene, V48, (E,Z)-undeca-1,3,5-triene, 49, 6-Butyl-1,4-cycloheptadiene, V50, 3,7-Dimethyldecane, V51, n-Butyl ether, V52, 1,3-Di-tert-butylbenzene, V53, 3,5-di-tert-butylphenol.

**Table 1 foods-14-03474-t001:** Changes in free amino acids during the fermentation of pineapple juice with *Ls. casei* LK-1.

Free Amino Acid (mg/100 mL)	Fermentation Time (h)					
	0	6	12	18	24	30
Cysteine (Cys)	0.57 ± 0.35 ^a^	0.63 ± 0.59 ^a^	0.37 ± 0.15 ^a^	0.48 ± 0.46 ^a^	0.47 ± 0.38 ^a^	0.33 ± 0.15 ^a^
Gamma-aminobutyric acid (GABA)	1.40 ± 0.10 ^a^	1.63 ± 0.26 ^a^	1.43 ± 0.16 ^a^	1.40 ± 0.10 ^a^	1.30 ± 0.10 ^a^	1.20 ± 0.30 ^a^
Ornithine (Orn)	0.37 ± 0.06 ^a^	0.37 ± 0.06 ^a^	0.43 ± 0.06 ^a^	0.43 ± 0.06 ^a^	0.43 ± 0.06 ^a^	0.37 ± 0.06 ^a^
Arginine (Arg)	2.20 ± 0.30 ^b^	2.70 ± 0.26 ^ab^	2.73 ± 0.40 ^ab^	3.03 ± 0.12 ^ab^	3.33 ± 0.49 ^a^	3.10 ± 0.10 ^a^
Aspartic Acid (Asp)	4.60 ± 0.30 ^d^	6.00 ± 0.20 ^bc^	5.87 ± 0.29 ^c^	6.43 ± 0.06 ^abc^	6.77 ± 0.47 ^ab^	7.27 ± 0.40 ^a^
Proline (Pro)	7.50 ± 1.61 ^b^	7.03 ± 0.67 ^b^	9.83 ± 0.50 ^b^	19.57 ± 0.67 ^a^	19.77 ± 3.07 ^a^	22.73 ± 1.06 ^a^
Tyrosine (Tyr)	3.17 ± 0.46 ^ab^	4.07 ± 0.21 ^a^	3.37 ± 0.42 ^ab^	3.53 ± 0.76 ^ab^	2.70 ± 0.69 ^ab^	2.63 ± 0.25 ^b^
Serine (Ser)	10.29 ± 0.42 ^a^	10.90 ± 0.26 ^a^	7.43 ± 0.15 ^b^	4.43 ± 0.76 ^c^	2.53 ± 0.40 ^d^	1.97 ± 0.15 ^d^
Asparagine (Asn)	40.67 ± 1.53 ^a^	40.23 ± 0.38 ^a^	33.37 ± 0.47 ^b^	30.23 ± 2.40 ^b^	25.73 ± 1.93 ^c^	24.50 ± 1.06 ^c^
Glutamine (Gln)	7.19 ± 0.16 ^ab^	7.47 ± 0.71 ^a^	6.27 ± 0.85 ^ab^	5.30 ± 0.30 ^bc^	3.23 ± 1.42 ^cd^	2.97 ± 0.57 ^d^
Glycine (Gly)	2.50 ± 0.26 ^ab^	2.57 ± 0.06 ^ab^	2.63 ± 0.06 ^a^	2.70 ± 0.10 ^a^	2.27 ± 0.12 ^b^	2.57 ± 0.17 ^ab^
Alanine (Ala)	7.83 ± 0.65 ^ab^	8.97 ± 0.21 ^a^	7.17 ± 0.50 ^bc^	5.67 ± 1.00 ^cd^	4.33 ± 0.75 ^de^	3.63 ± 0.15 ^e^
Valine (Val)	5.80 ± 0.62 ^a^	5.37 ± 0.71 ^a^	4.93 ± 0.42 ^a^	4.70 ± 0.70 ^a^	3.03 ± 0.47 ^b^	2.27 ± 0.59 ^b^
Methionine (Met)	3.30 ± 0.50 ^ab^	3.67 ± 0.21 ^a^	2.97 ± 0.12 ^abc^	2.90 ± 0.20 ^bc^	2.40 ± 0.10 ^c^	2.33 ± 0.21 ^c^
Isoleucine (Ile)	1.53 ± 0.15 ^ab^	1.60 ± 0.35 ^a^	1.27 ± 0.12 ^abc^	0.87 ± 0.12 ^c^	0.97 ± 0.12 ^bc^	1.13 ± 0.32 ^abc^
Leucine (Leu)	1.37 ± 0.06 ^b^	1.90 ± 0.00 ^a^	1.20 ± 0.26 ^b^	0.43 ± 0.12 ^c^	0.33 ± 0.06 ^c^	0.27 ± 0.21 ^c^
Threonine (Thr)	1.97 ± 0.23 ^a^	1.97 ± 0.21 ^a^	1.33 ± 0.15 ^ab^	1.20 ± 0.30 ^b^	0.90 ± 0.35 ^b^	0.97 ± 0.15 ^b^
Phenylalanine (Phe)	1.50 ± 0.35 ^a^	2.10 ± 0.00 ^a^	1.97 ± 0.29 ^a^	2.93 ± 0.95 ^a^	2.50 ± 0.52 ^a^	1.70 ± 0.70 ^a^
Lysine (Lys)	2.73 ± 0.12 ^a^	2.40 ± 0.00 ^b^	0.80 ± 0.00 ^c^	0.33 ± 0.06 ^d^	0.23 ± 0.06 ^d^	0.23 ± 0.06 ^d^

Results are presented as mean ± SD (*n* = 3). Values in the same row with different superscript lowercase letters are significantly different (*p* < 0.05).

**Table 2 foods-14-03474-t002:** 53 kinds of identified volatile compounds in the non-fermented and fermented pineapple juice (*n* = 3).

Volatile Compounds	Non-Fermented Pineapple Juice (ug/100 mL)	Fermented Pineapple Juice (ug/100 mL)
Esters		
[V1] Methyl 2-methylbutyrate	344.16 ± 8.18 ^a^	225.10 ± 22.49 ^a^
[V2] Ethyl butyrate	99.26 ± 11.20 ^a^	39.70 ± 1.72 ^b^
[V3] Ethyl 2-methylbutyrate	252.02 ± 10.04 ^a^	217.33 ± 3.91 ^b^
[V4] Isoamyl acetate	103.08 ± 1.35 ^a^	85.60 ± 4.17 ^b^
[V5] Ethyl valerate	5.24 ± 0.76 ^a^	2.74 ± 0.87 ^b^
[V6] Methyl caproate	899.04 ± 15.19 ^a^	844.70 ± 28.60 ^b^
[V7] Methyl (E)-hex-3-enoate	4.68 ± 0.25 ^a^	1.64 ± 0.21 ^b^
[V8] Ethyl caproate	829.13 ± 25.84 ^a^	549.96 ± 34.92 ^b^
[V9] Ethyl 3-hexenoate	26.20 ± 1.64 ^a^	13.67 ± 0.97 ^b^
[V10] Methyl 3-(methylthio)propionate	345.28 ± 46.18 ^a^	337.19 ± 19.66 ^a^
[V11] tert-Butyl propionate	1.45 ± 0.22 ^a^	1.61 ± 0.10 ^a^
[V12] Methyl 2-methylacetoacetate	9.39 ± 0.19 ^a^	5.75 ± 0.53 ^b^
[V13] Ethyl 3-methylthiopropionate	224.28 ± 13.10 ^a^	193.64 ± 13.16 ^b^
[V14] 4-Octenoic acid methyl ester	33.00 ± 5.47 ^a^	29.96 ± 7.09 ^a^
[V15] Methyl caprylate	80.18 ± 6.39 ^a^	52.37 ± 4.94 ^b^
[V16] Ethyl (Z)-oct-4-enoate	20.29 ± 2.19 ^a^	14.83 ± 3.02 ^b^
[V17] Ethyl caprylate	129.25 ± 7.42 ^a^	63.75 ± 3.77 ^b^
[V18] Methyl 3-acetoxyhexanoate	11.18 ± 3.87 ^a^	6.88 ± 0.94 ^a^
[V19] Methyl (Z)-4-decenoate	4.55 ± 0.83 ^a^	5.38 ± 0.29 ^a^
[V20] 1-Octen-3-yl-acetate	0	6.73 ± 0.04
[V21] Decanoic acid, ethyl ester	20.50 ± 3.22	0
[V22] Heptanoic acid, ethyl ester	0	7.93 ± 1.32
Terpenoids		
[V23] beta-(Z)-Ocimene	5.76 ± 0.25 ^a^	5.79 ± 0.49 ^a^
[V24] (E)-3,7-dimethylocta-1,3,6-triene	39.97 ± 3.43 ^a^	34.88 ± 2.77 ^a^
[V25] Caryophyllene	9.18 ± 0.92 ^a^	8.64 ± 0.54 ^a^
[V26] Guaiene	3.09 ± 0.31 ^b^	5.04 ± 0.25 ^a^
[V27] Copaen	105.62 ± 20.09 ^a^	108.93 ± 9.06 ^a^
[V28] beta-ylangene	11.26 ± 0.72 ^a^	11.33 ± 0.54 ^a^
[V29] Germacrene D	2.50 ± 0.36 ^a^	2.79 ± 0.058 ^a^
[V30] (+)-Sativene	13.26 ± 0.63 ^a^	12.92 ± 0.65 ^a^
[V31] (−)-Clovene	5.43 ± 0.32 ^a^	5.34 ± 0.19 ^a^
[V32] Valencene	24.58 ± 1.07 ^a^	25.52 ± 1.21 ^a^
[V33] β-Selinene	4.71 ± 0.54 ^a^	4.08 ± 0.28 ^a^
[V34] α-Muurolene	34.21 ± 2.42 ^b^	42.83 ± 2.37 ^a^
[V35] delta-Cadinene	10.23 ± 1.25 ^b^	14.15 ± 0.76 ^a^
[V36] Guaiazulene	1.32 ± 0.26 ^a^	1.78 ± 0.17 ^a^
[V37] (−)-Aristolene	82.61 ± 5.59 ^a^	83.59 ± 5.04 ^a^
Ketones		
[V38] 2-Heptanone	0	13.72 ± 2.96
[V39] 2-Nonanone	0	102.50 ± 19.67
[V40] 2-Undecanone	0	24.67 ± 7.97
[V41] 2,5-Dimethyl-4-methoxy-3(2H)-furanone	20.20 ± 2.03 ^a^	12.21 ± 1.47 ^b^
Acid		
[V42] Hexanoic acid	1.11 ± 0.12 ^b^	2.87 ± 0.25 ^a^
Aldehyde		
[V43] 2-Furaldehyde	16.40 ± 1.09 ^a^	10.27 ± 0.42 ^b^
Alcohols		
[V44] trans-Geraniol	10.32 ± 0.25 ^a^	7.33 ± 0.68 ^b^
[V45] 1,6-Hexanediol	1.61 ± 0.64	0
[V46] 1,2,3-Butanetriol	7.07 ± 2.27 ^b^	27.48 ± 4.10 ^a^
Others		
[V47] (3E,5E)-2,6-Dimethyl-1,3,5,7-octatetrene	22.99 ± 1.54 ^a^	14.80 ± 0.40 ^b^
[V48] (E,Z)-undeca-1,3,5-triene	28.10 ± 3.32 ^b^	33.64 ± 0.26 ^a^
[V49] 6-Butyl-1,4-cycloheptadiene	26.64 ± 3.95 ^a^	22.86 ± 1.74 ^a^
[V50] 3,7-Dimethyldecane	10.21 ± 1.69 ^a^	3.25 ± 0.18 ^b^
[V51] n-Butyl ether	2.45 ± 0.12 ^a^	2.14 ± 0.16 ^b^
[V52] 1,3-Di-tert-butylbenzene	24.39 ± 1.63 ^a^	22.73 ± 0.66 ^a^
[V53] 3,5-di-tert-butylphenol	3.50 ± 1.27 ^a^	3.56 ± 0.40 ^a^

Values are presented as mean ± SD (*n* = 3). Different lowercase letters within the same row indicate significant differences between non-fermented and fermented pineapple juice (*p* < 0.05).

## Data Availability

The original contributions presented in this study are included in the article. Further inquiries can be directed to the corresponding authors.

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
