# Peer review of "Effects of Lacticaseibacillus casei LK-1 Fermentation on Physicochemical Properties, Chemical Compositions, Antioxidant Activities and Volatile Profiles of Pineapple Juice"

_foods, 2025, doi:10.3390/foods14203474_

Round 1
Reviewer 1 Report
Comments and Suggestions for Authors
foods-3897579-peer-review-v1
The manuscript represent interesting results suggesting application of Lacticaseibacillus casei strain as starter culture in fermentation of pineapple juice. One important point, authors have mentioned that several probiotics properties of mentioned strain were previously studied. However, will be safe to mention “potential probiotic properties”, since according to international standards, strain can be calling probiotic only if appropriate animal and human studies have confirmed beneficial properties. Moreover, into fruit juice the strain is present, however, will be present into the entire shelf life period in a numbers recommended by the WHO? Thus, maybe using terms as “beneficia” or “potential probiotics” will be more appropriate.
Paper is presenting good set of experiments and deserve attention form the Editor; however, some adjustments and corrections needs to be made by the authors.
Please, in entire text be sure that microbial names are in italics. Please, see Ln19 and etc. Moreover, abbreviation of Lacticaseibacillus needs to be according to recommendations from Todorov SD, Baretto Penna AL, Venema K, Holzapfel WH, Chikindas ML. Recommendations for the use of standardized abbreviations for the former Lactobacillus genera, reclassified in the year 2020. Benef Microbes. 2023 Dec 12;15(1):1-4. doi: 10.1163/18762891-20230114. PMID: 38350480.
Ln40: Talking about origin of pineapple, will be maybe more appropriate to say, "South America".
Ln47, Please, Gram needs to be with capital, since is refers to name of Hans Gram.
Ln53: Please, correct Lactiplantibacillus plantarum
Ln55: Please, in this context, word "lactobacilli" need to be written without italics and without capital L. Since in this context referring to the English word.
Ln60-62: Please, provide references about the statement mentioned. Presume can be same as reference 10.
Ln71: Please explain NFC. It’s not fibrous carbohydrates? In fact, all abbreviations needs to be explained when for the first time appearing into the text.
If you have obtained the juice as commercial product, then, please, can you provide chemical composition of the product, taken from the product description. Any preservatives were used in the production of this juice?
Please, for all suppliers of material and equipment, provide address of the supplier, including name of the company, city, state (in case of federal countries) in abbreviated form and name of the country. In following occasions, only name of the country will be sufficient. Please, try to use address of the headquarters and not that of the local distributors.
Ln84: Strain identification do not need to be in italics.
Ln85: In this and following occasions, centrifugations needs to be in g force and and as rpm.
Ln90: in my opinion, do not need to specify how much NaOH was added, just specify the molarity of the NaOH used for the correction of the pH.
What original pH of the juice before correction of pH?
Before doing the determination of CFU/ml, authors have done test for the sterility of the applied and pasteurized juice? Please, on this and other occasions, specify numbers of the repetitions. Moreover, experiments under 2.4. will be beneficial if you can be provided with a bit more details. After enumeration have you tested if growing colonies are really representatives of studied culture?
Please, provide details on HPLC conditions and applied controls.
Section 2.8. needs to be with more details.
Section 2.9. need to be present with more details.
Please, on Ln162 change to Results and Discussion
Ln175: In my opinion word "probiotic" is not more appropriate. In this case this is starter culture.
Presence of growing bacterial culture will affect turbidity of the juice. Any comments, results on this?
Please, be sure that all results will be discussed appropriately. Discussion will need a more focus attention form the authors, as in the present way is very basic and episodically applied only for some of the obtained results.
Have you evaluated if the increase of lactic acid is related only to L lactic acid, or D lactic acid was cumulated as well? Please, provide this information.
In figure 51, use two different scales for concentration in control and experimental parts, so, the control parts can be better illustrated.
Please, abbreviations mentioned in the figures or tables need to be explained in the legends.
References needs to be formatted according to the instructions form Publisher and the Journal.
Author Response
Comments 1
Please, in entire text be sure that microbial names are in italics. Please, see Ln19 and etc. Moreover, abbreviation of Lacticaseibacillus needs to be according to recommendations from Todorov SD, Baretto Penna AL, Venema K, Holzapfel WH, Chikindas ML. Recommendations for the use of standardized abbreviations for the former Lactobacillus genera, reclassified in the year 2020. Benef Microbes. 2023 Dec 12;15(1):1-4. doi: 10.1163/18762891-20230114. PMID: 38350480.
Response 1
We sincerely thank the reviewer for this valuable reminder and for directing us to the essential reference. We have now carefully addressed both points as follows:
â‘ Italics for microbial names: We have thoroughly checked the entire manuscript and ensured that all microbial genus and species names (including the one on Line 19) are now presented in italics.
â‘¡Standardized abbreviation for Lacticaseibacillus: Following the recommendations outlined in the cited paper by Todorov et al. (2023), we have adopted the standardized abbreviation "Ls." for the genus Lacticaseibacillus. Specifically, Lacticaseibacillus casei is now abbreviated as "Ls. casei" and this abbreviated form is used consistently throughout the revised manuscript, with the exception of the reference section.
Comments 2
Ln40: Talking about origin of pineapple, will be maybe more appropriate to say, "South America".
Response2
We sincerely thank the reviewer for this insightful and accurate comment. Referring broadly to "South America" is more precise in the context of describing its geographical origin, and we agree with the reviewer. We have revised the manuscript accordingly on line 40.
Comments 3
Ln47, Please, Gram needs to be with capital, since is refers to name of Hans Gram.
Response 3
Thank the reviewer for the correction. In the revised version , we have changed "gram" to "Gram" (line 49).
Comments 4
Ln53: Please, correct Lactiplantibacillus plantarum.
Response 4
We were really sorry for our careless mistakes. Thank the reviewer for your reminder. In the revised version, we have changed "Lacticaseibacillus" to "Lactiplantibacillus"(line 56).
Comments 5
Ln55: Please, in this context, word "lactobacilli" need to be written without italics and without capital L. Since in this context referring to the English word.
Response 5
Thanks for your careful checks. Based on the comments from another reviewer, we have integrated the content from Ln 56-57 of the unrevised version into Ln 49-51 of the revised manuscript to avoid redundant expressions. Following this integration, we decided to remove the statement "L. casei is a commonly used Lactobacilli in food fermentation processes for enhancing antioxidant capacity, bioactivity, flavor, and quality" from the manuscript.
Comments 6
Ln60-62: Please, provide references about the statement mentioned. Presume can be same as reference 10.
Response 6
We sincerely appreciate the valuable comments. We have checked the literature carefully and added the reference into the part of Ln66 in the revised manuscript.
Comments 7
Ln71: Please explain NFC. It’s not fibrous carbohydrates? In fact, all abbreviations needs to be explained when for the first time appearing into the text.
Response 7
We sincerely thank you for your reminder. NFC denotes "Not From Concentrate." We regret the oversight and have now included its explanation in Section 2.1 (Ln84 of the revised manuscript). Furthermore, we have carefully reviewed the entire manuscript to ensure that all abbreviations (e.g., HPLC, CFU, PCA, etc.) are defined upon their first occurrence in the text.
Comments 8
If you have obtained the juice as commercial product, then, please, can you provide chemical composition of the product, taken from the product description. Any preservatives were used in the production of this juice?
Response 8
We thank the reviewer for valuable comments. As suggested , we have supplemented the nutritional composition data of the NFC pineapple juice and clarified in the revised manuscript ( Lines 78-80) that its ingredient list contains only 100% pineapple juice, with no additional components.
Comments 9
Please, for all suppliers of material and equipment, provide address of the supplier, including name of the company, city, state (in case of federal countries) in abbreviated form and name of the country. In following occasions, only name of the country will be sufficient. Please, try to use address of the headquarters and not that of the local distributors.
Response 9
We are grateful to the reviewer for pointing this out and apologize for our oversight. We have carefully checked the manuscript and supplemented all missing address and company information for the equipment and reagents in the the revised manuscript.
Comments 10
Ln84: Strain identification do not need to be in italics.
Response 10
Thank you for your reminder. It has been corrected in the revised version.The italics formatting for the strain identification has been corrected in the revised manuscript (Line 102).
Comments 11
Ln85: In this and following occasions, centrifugations needs to be in g force and and as rpm.
Response 11
Thank you for your valuable comment. We have revised the manuscript to expressthe centrifugal speeds in gravitational force (× g).
Comments 12
Ln90: in my opinion, do not need to specify how much NaOH was added, just specify the molarity of the NaOH used for the correction of the pH.What original pH of the juice before correction of pH?
Response 12
Thank you for this suggestion. We have revised the text accordingly.The original pH of the pineapple juice prior to adjustment was 3.8, and this information has been included in the manuscript It now reads: "The juice pH was adjusted from 3.8 to 6.8 with 3 M NaOH solution and pasteurized it at 80℃ for 10 min. " (Line 108).
Comments 13
Before doing the determination of CFU/ml, authors have done test for the sterility of the applied and pasteurized juice? Please, on this and other occasions, specify numbers of the repetitions. Moreover, experiments under 2.4. will be beneficial if you can be provided with a bit more details. After enumeration have you tested if growing colonies are really representatives of studied culture?
Response 13
Thank you for these valuable comments and for highlighting the need for greater methodological clarity. Please find our responses below:
(1) Regarding the sterility testing of the applied and pasteurized juice:
Yes, the sterility of the pasteurized juices prior to inoculation was confirmed by the absence of microbial growth in aliquots incubated for 48 h at 37℃. This detail has been added to the revised manuscript in Section 2.3(Lines 110-112).
(2) Regarding the specification of the number of repetitions:
We have now explicitly stated the number of the repetitions for all experiments in Section 2.3 and other section (e.g., "The entire procedure, from pretreatment to analysis, was carried out with three independent replicates.").
(3) Regarding providing more details for experiments in Section 2.4:
We have expanded the description of the methodology in Section 2.4 to provide a more comprehensive and stepwise account of the procedures.
(4) Regarding the confirmation of representative colonies:
To exclude the possibility of contamination, we conducted confirmatory analyses through morphological assessment of the strain using Gram staining and scanning electron microscopy (SEM). Additionally, the identity of the isolates was verified by comparing their colonial morphology with that of the original pure culture. These representative tests reinforce the reliability of our results. However, given that these methodological details pertain primarily to quality assurance rather than the central focus of this study, they have been omitted from the main manuscript to maintain narrative coherence and conciseness.The following pictures are our verification results.
Comments 14
Please, provide details on HPLC conditions and applied controls.
Response 14
Thank you for the reminder. We have substantially expanded the HPLC methodology in Section 2.7 to include detailed information on column temperature (30°C), the elution program, sample processing, and the total run time (30 min).
Comments 15
Section 2.8. needs to be with more details.
Response 15
Thank you for your valuable comment. We have revised Section 2.8 to provide a more detailed description of the procedure for the determination of free amino acids. The updated section now includes specific details regarding the model and specifications of the chromatographic column used, as well as the detection wavelength.
e.g.
“The separation was achieved using lithium citrate buffer system as mobile phase, employing a multi-step linear gradient program at a flow rate of 0.3 mL/min. The column temperature was programmed as follows: initially set at 40°C , increased to 70°C over 15 minutes, and then held constant. The injection volume was 20 μL. The absorbance at 440 and 570 nm was monitored after postcolumn reaction with ninhydrin reagent at 130°C.”
Comments 16
Section 2.9. need to be present with more details.
Response 16
Thank you for your valuable comment.We have provided more detailed descriptions of the experimental procedures for both the FRAP and DPPH assays in the revised manuscript. Specifically, we have included the exact volumes of reagents used, incubation conditions, and the instrument employed for absorbance measurement in the FRAP assay. For the DPPH assay, we have clarified the concentration of the DPPH solution and the sample preparation step. We believe these additions enhance the reproducibility and clarity of the methods.
Comments 17
Please, on Ln162 change to Results and Discussion.
Response 17
We sincerely appreciate the reviewer's thoughtful suggestion. As suggested, we have revised the section title from "Results" to "Results and Discussion" to better reflect the integrated content of this section (See line 228 ).
Comments 18
Ln175: In my opinion word "probiotic" is not more appropriate. In this case this is starter culture. Presence of growing bacterial culture will affect turbidity of the juice. Any comments, results on this?
Response 18
Thank you for your valuable comment regarding the use of the term "probiotic" in our manuscript. We agree with your observation that "starter culture" is indeed more scientifically appropriate in the context of our study. We have revised the terminology accordingly in the revised manuscript (Line 217).
Regarding your second point about the potential impact of bacterial growth on juice turbidity, we appreciate this insightful suggestion. While we did not systematically monitor turbidity changes in the current study, we fully acknowledge its relevance as an important physical property in fermented beverages. The conclusion section of the new version, we have included a discussion of this limitation in the revised manuscript and will consider incorporating turbidity measurements in our future investigations to comprehensively evaluate the physicochemical changes during fermentation .
Comments 19
Please, be sure that all results will be discussed appropriately. Discussion will need a more focus attention form the authors, as in the present way is very basic and episodically applied only for some of the obtained results.
Response 19
We sincerely appreciate your valuable suggestions. In the revised manuscript, we have incorporated more comprehensive discussions on key aspects, including changes in color parameters, antioxidant activity, and volatile components. The newly added content has been highlighted in yellow throughout the text to facilitate easy identification and reference.
Comments 20
Have you evaluated if the increase of lactic acid is related only to L lactic acid, or D lactic acid was cumulated as well? Please, provide this information.
Response 20
We sincerely thank the reviewer for this valuable question, which helps to improve the precision and quality of our work.The reviewer raises an important point regarding the stereoisomers of lactic acid. In this study, we quantified total lactic acid and did not perform a separate analysis to distinguish between the D- and L- forms. We acknowledge that this is a limitation of our current work.
Based on the microbial strains (Lactobacillus bulgaricus and Streptococcus thermophilus) used in our fermentation process, which are known to predominantly produce the L-(+)-lactic acid isomer, we hypothesize that the observed increase is primarily due to L-lactic acid. However, we fully agree that a specific analysis is necessary for confirmation. We have added a statement in the revised manuscript to clarify this point and will prioritize the specific quantification of D- and L-lactic acid isomers in our subsequent, more detailed investigations.
Comments 21
In figure 51, use two different scales for concentration in control and experimental parts, so, the control parts can be better illustrated.
Response 21
Thank you for your valuable comment. The issue you raised regarding the figure is crucial. We fully agree that optimizing the scale is necessary to better illustrate the data details of the control group. As you suggested, we have re-plotted Figure 51 using two different Y-axis scales for the control and experimental data, respectively. The revised chart is shown in the attachment.
Comments 22
Please, abbreviations mentioned in the figures or tables need to be explained in the legends.
Response 22
Thank you for this comment. We have now revised the figure/table legends to include a full explanation of all the abbreviations used.
Comments 23
References needs to be formatted according to the instructions form Publisher and the Journal.
Response 23
Thank you for this important reminder. We sincerely apologize for the oversight. The reference list has now been carefully reformatted to strictly adhere to the guidelines provided by the Publisher and the Journal. We have verified each entry against the required citation style to ensure consistency and completeness (See Section References).
Reviewer 2 Report
Comments and Suggestions for Authors
I have reviewed the manuscript foods-3897579 titled “Effects of Lacticaseibacillus casei LK-1 fermentation on physicochemical properties, chemical compositions, antioxidant activities, and volatile profiles of pineapple juice” by Shaodan Peng and co-authors.
The manuscript investigates the fermentation of pineapple juice using Lactobacillus casei LK-1, a strain with probiotic potential. The study examines changes in physicochemical parameters, phenolic and flavonoid contents, carbohydrate and organic acid profiles, free amino acids, antioxidant activity, and volatile compounds during a 30-hour fermentation. The work highlights the potential of L. casei LK-1 to create antioxidant-rich, aromatic fermented beverages.
Here are some suggestions for the authors in order to improve the manuscript.
Objectives
The objective of the study is clearly stated, but authors could better emphasize what differentiates this study from previous works on LAB-fermented juices.
Novelty
There are similar studies for other fruit juices. The novelty of the study lies in the application of LK-1 to pineapple juice. The authors should highlight the novelty of their study.
Methodology
- 7. Determination of carbohydrates and organic acids
Specify which enzyme kit was used for the determination of sugars.
Please provide a reference for the method used, if it was described more thoroughly in previous study. If not, please add missing critical parameters (detector type, oven temperature, use of internal standard, run time, software, calibration procedures, and validation parameters). Was there any pretreatment of the samples before the analyses?
- 8. Determination of free amino acids
Line 129. 0.22 μm filter – please provide the material of filter and manufacturer.
Report the full names of the amino acids along with the abbreviations.
- 11. Statistical analysis
Please add the software used (version and manufacturer) for the analyses mentioned.
Results and discussion
The authors could discuss how the changes in volatiles may affect sensory characteristics and consumer’s acceptance.
The authors should acknowledge limitations of the study (eg. lack of sensory evaluation) and areas for future work.
Author Response
Comments 1
The objective of the study is clearly stated, but authors could better emphasize what differentiates this study from previous works on LAB-fermented juices.
Response 1
Thank you for your valuable suggestions. We fully agree with your perspective, as clearly delineating the distinctions between this study and existing literature is essential to underscore its innovative contributions. In accordance with your recommendations, we have revised the introduction section (e.g., Paragraph 4) to better highlight these aspects.
The primary distinctions of this research compared to prior studies are summarized in the following two areas:
- Differences in fermentation strains: The strain employed in this study is a potential probiotic strain—Lactobacillus casei K-1. Previous findings indicate that this strain exhibits strong tolerance to acid, salt, and sugar, along with robust acid production and growth capacity in fermented pitaya juice, making it a promising candidate for use in fermentation processes.
- Comprehensive investigation: This study presents a systematic evaluation of the effects of Lactobacillus casei K-1 on fermented pineapple juice, encompassing multiple parameters such as physicochemical properties, volatile compounds, color attributes, and chemical composition. In contrast, previous studies have predominantly focused on changes in physicochemical properties and antioxidant activities, with limited attention given to detailed chemical profiling.
These distinguishing features have been explicitly emphasized in the revised introduction to enhance clarity and contextual relevance.
Comments 2
There are similar studies for other fruit juices. The novelty of the study lies in the application of LK-1 to pineapple juice. The authors should highlight the novelty of their study.
Response 2
We thank the reviewer for pointing this out and for acknowledging the novelty of applying the LK-1 strain. In the revised manuscript, we have clarified the novelty highlighted in the cited literature, which primarily resides in the utilization of lactic acid bacterial fermentation to enhance the functional properties and quality of various food substrates. This provides a theoretical foundation for the application of the LK-1 strain.
Further, we have now further strengthened the argument for our study's novelty by emphasizing two key aspects:
â‘ The suitability of a robust strain and a challenging matrix: We highlight that this strain possesses unique resilience to high-sugar and high-acidity environments and is suitable for the fermentation of fruits with high acidity and sugar content, such as pineapple juice.
â‘¡The gap in comprehensive profiling for fermemtd pineapple juice: We clarify that despite existing studies on LAB and fruit juices, a comprehensive analysis of the effects of a Ls.casei strain on the multifaceted quality attributes of pineapple juice is still limited.
The novelty is now highlighted in the revised introduction:
"The strain has a short growth cycle, reaching the stationary phase within approximately 12 h of cultivation and had demonstrated good acid productivity, yielding up to 16 g/L of lactic acid during the fermentation of pitaya juice [11]. These attributes collectively indicate good fermentation potential, rendering the strain particularly suitable for the fermentation of high-sugar and acidic fruit juices, such as pineapple juice"(The third paragraph of the introduction)
“Although LAB have been investigated for fruit juice fermentation, a comprehensive analysis of the effects of the Ls. casei strain on the multifaceted quality attributes of pineapple juice—particularly on the dynamics in key chemical components such as sugars, organic acids, and free amino acids—is still limited. ”(The fourth paragraph of the introduction)
Comments 3
Specify which enzyme kit was used for the determination of sugars. Please provide a reference for the method used, if it was described more thoroughly in previous study. If not, please add missing critical parameters (detector type, oven temperature, use of internal standard, run time, software, calibration procedures, and validation parameters). Was there any pretreatment of the samples before the analyses?
Response 3
Thank you for your valuable comments. We have revised the manuscript to provide the detailed information regarding the analytical methods for sugars and organic acids. The specific revisions are as follows:
â‘ Regarding the determination of sugars:
We have clearly stated that the Y15 automatic analyzer (BioSystems) will be used in conjunction with the D-glucose/D-fructose and sucrose/glucose/fructose assay kits for analysis. At the same time, the key sample pretreatment steps (centrifugation, dilution) and determination parameters (detection wavelength 340 nm, reaction temperature 37°C) were supplemented.
â‘¡ Regarding the determination of organic acids:
We have cited previous studies (Xu et al. [15]) as the basis of this method and optimized it on this basis. Meanwhile, we have supplemented all the key parameters required by the reviewers:
Column temperature: The column temperature is maintained at 30°C.
Sample pretreatment: It clearly describes that after centrifugation and dilution, the sample is filtered using a 0.22 μm PES membrane filter.
Quantitative method: The external standard method was adopted for quantification.
Running time: The running time of the analytical method is 30 minutes.
Comments 4
Line 129. 0.22 μm filter—please provide the material of filter and manufacturer. Report the full names of the amino acids along with the abbreviations.
Response 4
Thank you for your valuable suggestions. The two points you raised are of significant importance and will greatly contribute to refining the experimental details, as well as enhancing the rigor and reproducibility of the manuscript. Below is our response to your comments, which has been incorporated into the revised version accordingly.
- Regarding the materials and manufacturers of the 0.22 μm filters
We appreciate your observation, which highlights an essential omission in the original submission. We sincerely apologize for the oversight in not including this critical information.
Material: The 0.22 μm filter used in this study is composed of PES.
Manufacturer: The filter is manufactured by Tianjin Jinteng Experimental Equipment Co., Ltd. (China).
In the revised manuscript, we have updated the relevant description in Section 2.8 of the "Materials and Methods" section (Line 174-176) as follows:
"... The supernatant was diluted 1:5 (v/v) and filtered through a 0.22 μm PES filter (Tianjin Jinteng Experimental Equipment Co., Ltd., Tianjin, China)"
- Regarding the full names and abbreviations of amino acids
We thank you for your suggestion to standardize the use of scientific nomenclature. In accordance with best practices, we have changed the original abbreviations to full letters in table 1 of Section 3.6 ("Results and Discussion") in the revised manuscript. Furthermore, in the main text, each amino acid is introduced with its full name followed by its abbreviation in parentheses at first mention, ensuring clarity and consistency throughout the document.
Comments 5
Please add the software used (version and manufacturer) for the analyses mentioned.
Response 5
Thank you for your reminder. In the revised manuscript, we have included the manufacturer and its location for the data analysis software in the section of Statistical analysis. For example, the data were analyzed using OriginPro 2024 SR1 (OriginLab Corporation, Northampton, MA, USA).
Comments 6
The authors could discuss how the changes in volatiles may affect sensory characteristics and consumer’s acceptance.
Response 6
We sincerely thank the reviewer for this valuable suggestion. In response, we have substantially revised the manuscript to include a more in-depth discussion on how the changes in volatile compounds may influence the sensory characteristics and consumer acceptance of the fermented pineapple juice (FPJ). The key additions and corresponding rationale are summarized as follows:
â‘ Regarding Esters: We noted that the content of most esters, which are primary contributors to the typical fruity and sweet aroma of pineapple juice, decreased significantly after fermentation. Specific examples with quantitative data were provided (e.g., ethyl caproate, isoamyl acetate, ethyl butyrate). We further discussed that this decline likely leads to a substantial attenuation of the intense fruity and sweet aroma profile, resulting in an overall milder and softer flavor.
â‘¡Regarding Terpenoids: We highlighted that most terpenoids remained stable, helping to maintain the fundamental floral and fruity aroma. Furthermore, the accumulation of specific terpenoids (e.g., guaiene, α-muurolene, delta-cadinene) with woody, spicy, and earthy notes was discussed as a factor that likely contributes to a more complex and potentially more mature aroma profile.
â‘¢Regarding Methyl Ketones and Aldehydes: The generation of methyl ketones (2 - heptanone, 2 - nonanone, 2 - undecanone) was discussed as potentially enriching the flavor with fruity, oily, and floral notes. Concurrently, the significant decrease in 2 - furaldehyde was explicitly linked to avoiding undesirable off - flavors, thereby positively aligning with preferred sensory profiles and likely benefiting consumer acceptance.
â‘£Regarding Hexanoic Acid: We specifically addressed the significant increase in hexanoic acid, noting its association with negative descriptors (cheesy, rancid). We explicitly stated that this elevation "poses a potential risk to sensory acceptability if its concentration exceeds the perceptual threshold," directly linking this chemical change to a potential negative impact on consumer preference.
Comments 7
The authors should acknowledge limitations of the study (eg. lack of sensory evaluation) and areas for future work.
Response 7
We thank the reviewer for this constructive suggestion. We fully agree that acknowledging the study's limitations and outlining future research directions will strengthen the manuscript. As per your advice, we have added a dedicated paragraph in the Conclusion section to explicitly address these points.
The newly added text (in the revised manuscript) is as follows:
"This study provides a comprehensive foundation for the application of the potential probiotic strain Ls. casei LK-1 in fruit juice fermentation. However, this study has limitations, primarily the lack of turbidity analysis, sensory evaluation to confirm the perceived quality improvements, and shelf-life testing to assess product stability. Future research should focus on conducting sensory analysis, evaluating in vivo bioactivity, and assessing the product's shelf life to facilitate the development of consumer-acceptable functional fermented beverages."
In addition, we have clarified the future development direction, which is expressed as follows.
“Future research should focus on conducting sensory analysis, evaluating in vivo bioactivity, and assessing the product's shelf life to facilitate the development of consumer-acceptable functional fermented beverages”
Reviewer 3 Report
Comments and Suggestions for Authors
Abstract
- Define the analytical methods used (e.g., DPPH, FRAP, GC-MS) rather than referring to measurements generically.
- Avoid repeating measurement categories (lines 20–24 and 25–31); consolidate to reduce redundancy.
- The phrase “undesirable aromas” should be replaced with precise descriptions or named compounds.
- Rephrase “highlighted the fermentation potential…” to a more objective statement of findings.
- Indicate whether observed differences are statistically significant.
Introduction
- Correct the sentence beginning “how to preserve or enhance…”, it is grammatically incorrect and should be rewritten as a clear objective or question.
- Avoid repeating general benefits of LAB in multiple paragraphs; consolidate for conciseness.
- Revise “L. casei are…” to “L. casei is…” for subject-verb agreement.
- Clarify the phrase “It found that L. casei…”; it lacks a clear subject. Revise for clarity.
- Replace “survey” with “study” or “investigation” in line 65, which is more accurate for experimental work.
- Provide more detailed rationale for selecting the LK-1 strain specifically, e.g., comparative benefits or previous application in fruit juice matrices.
Materials and Methods
- Clearly state the number of biological and technical replicates used for each assay.
- Specify if the control juice also underwent pasteurization and incubation to isolate the effects of fermentation alone.
- Justify the choice of 30-hour fermentation duration; was it optimized in pretests?
- Include catalog numbers or batch IDs for commercial kits and reagents where applicable.
- Clarify whether GC-MS calibration involved internal standards or library match thresholds for identification.
- The description of volatile quantification lacks units and clarity. Revise the formula to explicitly include all variables and assumptions.
- Confirm whether statistical assumptions (e.g., normality) were tested before applying ANOVA and Tukey tests.
- In section 2.3, clarify the basis for adjusting pH to 6.8 prior to fermentation. Was this optimized or standardized?
Results and Discussion
- Provide p-values or indicators of statistical significance for all major comparisons, not only in figures.
- Reassess the claim that color changes (e.g., L*, a*, b*) indicate “improvement” without sensory validation; such interpretations are subjective.
- The term “stable sweet amino acids” needs definition; is “stable” statistically defined?
- Avoid attributing changes in phenolic content or amino acids to enzymes or heat without direct measurement; indicate this as a hypothesis.
- Link changes in volatiles (e.g., reduction in 2-furaldehyde, increase in esters) to sensory implications only if validated with panel data; otherwise, qualify as potential effects.
- Refrain from using phrases like “enhanced antioxidant activity” unless the observed change is statistically significant and biologically relevant.
- Discuss the inconsistency between decreased total phenolics and increased antioxidant activity, provide a mechanistic hypothesis or acknowledge this gap.
- In the PCA results, report the percentage of variance explained by PC1 and PC2 to support the significance of group separation.
- Many discussions interpret changes as beneficial without objective support (e.g., “improved flavor profile”), these should be revised or attributed to literature comparisons.
- Condense figure captions that restate the main text, focus on interpreting figure content rather than repeating methods.
- Why are Figures 6 and 7 inconsistent in format when both use a bar diagram, and use the same style as Figure 7 for Figure 6.
Conclusion
- Avoid repeating detailed data points (e.g., “22.05 ± 0.94 g/L lactic acid”) in the conclusion; focus on general trends.
- Reword claims such as “highlighting strong acid-producing ability” unless compared to other strains or thresholds.
- Qualify statements about improved sensory properties or health value unless validated through sensory testing or bioactivity studies.
- Clearly identify study limitations, g., absence of sensory panels, shelf-life testing, or fermentation optimization.
- Include future directions that align with observed results (e.g., strain screening, multi-strain fermentation, consumer acceptability studies).
Author Response
Comment 1
Abstract: Define the analytical methods used (e.g., DPPH, FRAP, GC-MS) rather than referring to measurements generically.
Response 1
We strongly agree with this suggestion. The methods for assessing antioxidant activity and volatile compounds have now been explicitly defined in the revised text. Specifically, we have replaced "antioxidant activities" with "antioxidant activities (assessed by DPPH and the FRAP methods)", "volatile profiles" with "volatile profiles (analyzed by GC-MS)", and "organic acids" with "organic acids (analyzed by HPLC)" in the Abstract.
Comment 2
Abstract: Avoid repeating measurement categories (lines 20–24 and 25–31); consolidate to reduce redundancy.
Response 2
Thank you for sharing your valuable comments. In the revised version, we have replaced the original sentence, “The pH, total soluble solids, color parameters, phenolics and flavonoids, carbohydrates, organic acids, free amino acids, antioxidant activities, and volatile profiles of fermented pineapple juice were characterized.” with the following: “The strain growth, physicochemical parameters, phenolics and flavonoids, carbohydrates, organic acids (analyzed by HPLC), free amino acids (FAAs), antioxidant activities (assessed by DPPH and FRAP methods), and volatile profiles (analyzed by GC-MS) of fermented pineapple juice (FPJ) were characterized.”
Comment 3
Abstract: The phrase “undesirable aromas” should be replaced with precise descriptions or named compounds.
Response 3
Thank you for your valuable comments. As suggested, the subjective phrase "undesirable aromas" has been removed in the Abstract.
Comment 4
Abstract: Rephrase “highlighted the fermentation potential…” to a more objective statement of findings. Indicate whether observed differences are statistically significant.
Response 4
Thank you for your valuable comments. In response to your feedback, we have carefully revised the manuscript.
The revised version is presented as follows:
“These findings demonstrate the feasibility of using Ls. casei for fermentation to develop novel fermented pineapple juice with good antioxidant properties and a modified volatile profile.”
Comment 5
Abstract: Indicate whether observed differences are statistically significant.
Response 5
Thank you for your valuable feedback. In the revised manuscript, we have incorporated your suggestion by explicitly indicating the statistical significance of the key findings in the abstract. Specifically:
â‘ Changes in color parameters (L*, a*, b*), pH, and total soluble solids are now annotated with "p < 0.05", confirming that these differences are statistically significant.
â‘¡The reduction of the 2-formaldehyde is also reported with "p < 0.05", demonstrating that this decrease is statistically significant.
Comment 6
Introduction:Correct the sentence beginning “how to preserve or enhance…”, it is grammatically incorrect and should be rewritten as a clear objective or question.
Response 6
Thank you for pointing out the grammatical issue in the sentence beginning with “how to preserve or enhance…”. As suggested, we have rewritten the sentence to clearly state the objective. The revised sentence now reads:
“Therefore, developing strategies to preserve or enhance the sensory qualities, nutritional value, and bioactive components of pineapple juice is important.”
Comment 7
Introduction:Avoid repeating general benefits of LAB in multiple paragraphs; consolidate for conciseness.
Response 7
We sincerely thank the reviewer for this valuable suggestion. As recommended, we have consolidated the general benefits of lactic acid bacteria (LAB) and the specific advantages of Ls. casei fermentation into a more concise and focused narrative in the revised manuscript. Specifically, we have merged the previously separated descriptions of LAB’s general roles and the fermentation effects of Ls. casei into one coherent paragraph (second paragraph of the revised text), eliminating redundant statements and emphasizing only the most relevant information related to our study. This revision enhances the logical flow and conciseness of the introduction while maintaining the necessary background context.
Comment 8
Introduction:Revise “L. casei are…” to “L. casei is…” for subject-verb agreement.
Response 8
Thank you very much for your valuable suggestions regarding this manuscript. The subject-verb agreement issue you identified is highly significant, and we have carefully addressed it in the revised version accordingly.
The specific revisions are as follows:
Previously: "Ls. casei are the most prominent bacteria among LAB genera..."
Revised sentence: "Ls. casei is one of the most prominent bacteria among LAB species..."
Comment 9
Introduction:Clarify the phrase “It found that L. casei…”; it lacks a clear subject. Revise for clarity.
Response 9
We sincerely appreciate your valuable feedback regarding the manuscript. Your observation that the sentence "It found that L. casei..." lacks a clear subject is well taken, as this omission indeed compromises the sentence's clarity and academic rigor. We fully agree with your assessment. In response to your comments, we have thoroughly revised this section in the updated draft. The specific revision is as follows:
"In addition, previous research has found that Lactobacilli fermentation increased amino acids and ash contents, reduced fat content, and improved pasting properties in yellow pea flours [8]."
Comment 10
Introduction:Replace “survey” with “study” or “investigation” in line 65, which is more accurate for experimental work.
Response 10
We sincerely thank the reviewer for this valuable suggestion. As recommended, the word "survey" has been replaced with "investigation" in the revised manuscript (please refer to the last paragraph of the Introduction in the revised version) to more accurately describe our experimental work. We believe this change enhances the precision of our terminology.
Comment 11
Introduction:Provide more detailed rationale for selecting the LK-1 strain specifically, e.g., comparative benefits or previous application in fruit juice matrices.
Response 11
Thank you for this constructive suggestion. We have now provided a more detailed rationale for the selection of Ls. casei LK-1 in the revised manuscript. Specifically, we have clarified that this strain not only exhibits good acid and bile salt tolerance—maintaining high viability in simulated gastrointestinal conditions—but also demonstrates exceptional adaptability and robust growth in challenging food matrices. Furthermore, we have included additional data from previous studies showing that Ls. casei LK-1 possesses a short growth cycle, high acid productivity, and has been successfully applied in the fermentation of other high-sugar fruit juices, such as pitaya fruit juice, yielding up to 16 g/L of lactic acid. These combined traits underscore its suitability and potential advantages for fermenting acidic and high-sugar fruit juices like pineapple juice.
Comment 12
Materials and Methods:Clearly state the number of biological and technical replicates used for each assay.
Response 12
Thank you for your reminder. In the revised manuscript, we have stated the number of biological and technical replicates used for each assay, for example, The entire procedure, from pretreatment to analysis, was carried out with three independent replicates;All experiments were performed in triplicate.
Comment 13
Materials and Methods:Specify if the control juice also underwent pasteurization and incubation to isolate the effects of fermentation alone.
Response 13
We thank the reviewer for this valuable comment. We have revised the manuscript to clarify this point. As detailed in the revised Methods section 2.3, both the fermented and the non-fermented control juice underwent identical pretreatment, which included pH adjustment and pasteurization at 80 °C for 10 min.
Comment 14
Materials and Methods:Justify the choice of 30-hour fermentation duration; was it optimized in pretests?
Response 14
Thank you for your comment. We chose a 30-hour fermentation time based on the optimal conditions determined by the previous process optimization experiments. In the revised version, the references on which this fermentation condition is based have been supplemented.
References:
Lian, Y.; Shaodan, P.; Junjie, M.; Chenghui, Z.; Guang, W.; Xiaobing, H.; Jihua, L.; Liangkun, L. Fermentation characteristics of Lactobacillus casei LK-1 and its application in pineapple juice. Sci. Technol. Food Ind. 2024, 45, 199-207.
Comment 15
Materials and Methods:Include catalog numbers or batch IDs for commercial kits and reagents where applicable.
Response 15
Thank you for your valuable comment regarding the inclusion of catalog numbers for commercial kits and reagents. We have carefully revised the “Materials” section accordingly and have now provided the catalog/reference numbers for all applicable commercial kits and reagents, including the T-AOC assay kit, MRS agar, PCA, MRS broth, and the assay kits for D-glucose/D-fructose and sucrose/glucose/fructose.
The corresponding revisions can be found in the highlighted portions of the revised manuscript.
Comment 16
Materials and Methods:Clarify whether GC-MS calibration involved internal standards or library match thresholds for identification. The description of volatile quantification lacks units and clarity. Revise the formula to explicitly include all variables and assumptions.
Response 16
Thank you for your valuable suggestions. In response to your feedback, we have enhanced the manuscript by providing additional details and clarifications regarding the qualitative and quantitative methodologies employed in GC-MS analysis.
Specific revisions are outlined as follows:
â‘ With respect to the use of internal standards for quantification: We have explicitly specified the concentration unit (µg/100mL) in the formula used to calculate relative concentrations, thereby emphasizing that the quantification approach is based on an internal standard (octadecane).
The revised formula: Ci = (Cis × Ai)/Ais (Ci is the concentration of the analyte (μg/100 mL), Cis is the final concentration of the internal standard in vial (μg/100 mL), Ai is the chromatographic peak area of the analyte; Ais is the chromatographic peak area of the internal standard)
â‘¡Regarding the spectral library matching threshold: We have added clarification that the identification of volatile compounds was performed by comparing their mass spectra against the NIST14s/Wiley9 mass spectral libraries. To ensure the reliability and accuracy of compound identification, a minimum spectral match similarity threshold of 85% was applied.
Comment 17
Materials and Methods:Confirm whether statistical assumptions (e.g., normality) were tested before applying ANOVA and Tukey tests.
Response 17
Thank you for your valuable comment. As suggested, we have confirmed that the normality assumption was tested prior to performing one-way ANOVA and Tukey’s test. Specifically, the Shapiro-Wilk test was applied to verify the normal distribution of the data. The revised sentence in the manuscript (The section of Statistical analysis ) now reads:
“Data normality was confirmed using the Shapiro-Wilk test before further analysis.”
Comment 18
Materials and Methods:In section 2.3, clarify the basis for adjusting pH to 6.8 prior to fermentation. Was this optimized or standardized?
Response 18
We appreciate your feedback. The adjustment of the pH to 6.8 prior to fermentation was determined based on the findings of our previous process optimization studies. In the revised manuscript, we have provided additional clarification and incorporated relevant literature to support the rationale for this pH adjustment step.
References:
Lian, Y.; Shaodan, P.; Junjie, M.; Chenghui, Z.; Guang, W.; Xiaobing, H.; Jihua, L.; Liangkun, L. Fermentation characteristics of Lactobacillus casei LK-1 and its application in pineapple juice. Sci. Technol. Food Ind. 2024, 45, 199-207.
Comment 19
Results and Discussion:Provide p-values or indicators of statistical significance for all major comparisons, not only in figures.
Response 19
Thank you for your valuable feedback and thorough review of our manuscript. We appreciate your comment regarding the need to provide p-values or indicators of statistical significance for all major comparisons in the text. This is a crucial point, and we fully agree that it enhances the clarity and rigor of our results presentation.
In response to your suggestion, we have thoroughly revised the relevant paragraphs in the Results section as follows:
For all primary inter-group comparisons (e.g., Experimental Group vs. Control Group), we now explicitly report the exact p-values (e.g., p <0.05) within the main text.
For comparisons that were not statistically significant, we have also stated "no statistically significant difference" (p > 0.05) in the text.
These changes have been implemented throughout the Results section of our manuscript,for example,“The viable cell count of Ls. casei LK-1 in pineapple juice increased rapidly within the first 12 h of fermentation, with no significant (pï¹¥0.05) alterations observed thereafter (Fig. 2A). During the 30 h fermentation period, the bacterial count increased notably from 6.72 ± 0.05 log CFU/mL to 9.07 ± 0.07 log CFU/mL (p < 0.05)......”
Comment 20
Reassess the claim that color changes (e.g., L*, a*, b*) indicate “improvement” without sensory validation; such interpretations are subjective.
Response 20
Thank you for your valuable feedback. In the revised manuscript, we have replaced the term "...thereby improving its color profile. " with " This color shift towards a brighter, more yellow appearance is often associated with enhanced visual quality and consumer appeal [23]".
Comment 21
The term “stable sweet amino acids” needs definition; is “stable” statistically defined?
Response 21
We sincerely thank the reviewer for this valuable comment. We apologize that this issue arises from insufficient precision in our selection of terminology. In the revised manuscript, we have addressed this concern by replacing the subjective term “stable” with a more precise statistical description. Specifically, we have amended the sentence on Lines 362-364, to now read:
“However, a significant increase in Proline (Pro) levels resulted in no statistically significant net change (p > 0.05) in the total content of sweet free amino acids between 0 h and 30 h.”
Comment 22
Avoid attributing changes in phenolic content or amino acids to enzymes or heat without direct measurement; indicate this as a hypothesis.
Response 22:
We sincerely appreciate your valuable feedback on this manuscript. We have revised the section addressing the changes in phenolic and flavonoid content. Specifically, we have removed the original statements that directly attributed the reduction of these compounds to "thermal processing" or "enzyme activity" and instead introduced more cautious, hypothesis-driven explanations that underscore their tentative nature.
The revised text now states:
"Many reports have indicated that potential enzymes (e.g., glycosidases, polyphenol oxidases) produced by LAB could promote the degradation or transformation of phenolic compounds [12,26]. Therefore, it is hypothesized that the observed decrease in TPC and TFC could be attributed to the transformation of phenolic compounds by enzymes produced by the fermenting strain."
The original statements :
“The reduction in phenolic and flavonoid components might be attributed to thermal processing (37 ℃) and the enzymes produced by LAB. ”
Comment 23
Link changes in volatiles (e.g., reduction in 2-furaldehyde, increase in esters) to sensory implications only if validated with panel data; otherwise, qualify as potential effects.
Response 23
We sincerely thank you for your reminder. Upon verification, we confirmed that several similar issues were present in the manuscript. In the revised version, all instances of statements regarding the impact of changes in volatile components on product quality have been consistently revised to "possible impact" or "potential impact" to more accurately reflect the rigor of scientific inference. Detailed modifications are clearly indicated in gray within Section 3.8 of the revised version.
Comment 24
Refrain from using phrases like “enhanced antioxidant activity” unless the observed change is statistically significant and biologically relevant.
Response 24
Thank you for your valuable feedback. In the revised version, we have refrained from using absolute language. For instance, we have change "This increase in antioxidant... " to "This increase in DPPH radical scavenging capability and FRAP ..." in the Section 3.7.
Comment 25
Discuss the inconsistency between decreased total phenolics and increased antioxidant activity, provide a mechanistic hypothesis or acknowledge this gap.
Response 25
Thank you for your valuable feedback. The observed inconsistency between the decrease in total phenolics and the increase in antioxidant activity is indeed interesting and warrants further discussion. We have revised the manuscript to address this point. As suggested, we have added a mechanistic hypothesis to explain this apparent discrepancy. Specifically, we propose that during fermentation, microbial enzymes (such as β-glucosidase) may break down complex phenolic compounds (like glycosides) into simpler, more bioaccessible forms, such as free aglycones or phenolic acids. While this transformation could lead to a decrease in the measured total phenolics (as some conjugated forms might not be fully detected by the assay), the released free forms often possess higher antioxidant potency. Additionally, the release of other non-phenolic antioxidants, such as certain peptides or Maillard reaction products generated during fermentation, might also contribute to the enhanced antioxidant activity.
This explanation has been incorporated into the revised discussion section (please refer to the modified text). We agree that the precise mechanisms and the relative contribution of different factors require further experimental validation, and we have acknowledged this in the text.
Comment 27
In the PCA results, report the percentage of variance explained by PC1 and PC2 to support the significance of group separation.
Response 27
Thank you for your valuable comment regarding the PCA results. As suggested, we have revised the manuscript to include the percentage of variance explained by the first two principal components (PC1 and PC2). Specifically, we now state that PC1 and PC2 cumulatively accounted for 80.9% of the total variance (PC1: 64.0%, PC2: 16.9%). This addition provides quantitative support for the distinct separation observed between the fermented and control groups, reinforcing the significance of the fermentation effect on the volatile profiles.
Comment 28
Many discussions interpret changes as beneficial without objective support (e.g., “improved flavor profile”), these should be revised or attributed to literature comparisons.
Response 28
Thank you for your valuable suggestions. In the revised draft, we have carefully reviewed the entire text and made revisions one by one for similar issues. All the modified contents have been marked in gray font for easy reference.
Comment 29
Condense figure captions that restate the main text, focus on interpreting figure content rather than repeating methods.
Response 29
Thank you very much for your valuable suggestions. We fully agree with your view that the captions to figures should avoid repeating the methods already detailed in the main text, and instead focus on interpreting the content of the figures and highlighting the key findings.Based on your suggestions, we have conducted a comprehensive review and revision of all the captions in the text. The specific revisions include:
â‘ Redundant method descriptions have been removed: experimental steps and process descriptions that have been clearly stated in the "Materials and Methods" or the main text of the results have been eliminated.
â‘¡Simplify the title
The following is an example of our modification to the caption of figure [8] to specifically illustrate the improvements we have made:
Before modification (original image caption) :
Figure [8]. Differentiation of non-fermented and fermented pineapple juice based on the volatile compound profile: (A) cluster heat map, (B) PCA. Fermented pineapple juice (FPJ) was fermented at 37 ℃ for 30 h, while non-fermented pineapple juice (NFPJ) was not added to the starter and incubated at 37 ℃ for 30 h.
Revised (New illustration note)
Figure [8]. [A] Volatile compound profiling to differentiate non-fermented and fermented pineapple juice: (A) cluster heatmap, (B) principal component analysis (PCA).
Comment 30
Why are Figures 6 and 7 inconsistent in format when both use a bar diagram, and use the same style as Figure 7 for Figure 6.
Response 30
We sincerely thank the reviewer for this insightful comment. We apologize for the inconsistency in the formatting between Figures 6 and 7 in our original manuscript. Following the reviewer's suggestion, we have now carefully revised Figure 6 to adopt the exact same visual style as Figure 7.
Comment 31
Conclusion: Avoid repeating detailed data points (e.g., “22.05 ± 0.94 g/L lactic acid”) in the conclusion; focus on general trends. Reword claims such as “highlighting strong acid-producing ability” unless compared to other strains or thresholds. Qualify statements about improved sensory properties or health value unless validated through sensory testing or bioactivity studies. Clearly identify study limitations, g., absence of sensory panels, shelf-life testing, or fermentation optimization. Include future directions that align with observed results (e.g., strain screening, multi-strain fermentation, consumer acceptability studies).
Response 31
Thank you very much for your valuable suggestions on this manuscript. We have comprehensively revised the conclusion section based on your suggestions. The specific explanations are as follows:
- Regarding avoiding repetitive detailed data points:
We have removed the specific value of lactic acid concentration "22.05 ± 0.94 g/L" and modified its description to "Concurrently, a substantial increase in lactic acid concentration was observed". To focus on the overall trend of a significant increase in lactic acid content.
- Regarding the restatement of assertions such as "strong acid production capacity" :
We have removed the expression "highlighting the strong acid-producing ability of L. casei LK-1" in the revised manuscript.
- Regarding claims that limit "sensory improvement" or "health value" :
We have removed the qualitative descriptions such as "antioxidant-rich and aromatic" in the original text that have not been directly verified through sensory or biological activity experiments. Instead, we adopt a more objective expression It is explained that fermentation brings "modified physicochemical properties, enhanced DPPH free radical scavenging activity and FRAP, and an altered volatile profile.
- Regarding the clarification of research limitations
We have added an independent paragraph at the end of the conclusion of the revised draft, clearly pointing out the limitations of this study: “However, this study has limitations, primarily the lack of sensory evaluation to confirm the perceived quality improvements and shelf-life testing to assess product stability.”
- Regarding proposing future research directions consistent with the results:
Also in the newly added paragraphs, we have proposed future work directly related to the "changes in physicochemical properties, enhanced antioxidant capacity, and alterations in the flavor substance spectrum" discovered in this study: “Future research should focus on conducting sensory analysis, evaluating in vivo bioactivity, and assessing the product's shelf life to facilitate the development of consumer-acceptable functional fermented "beverages." These directions aim to verify their sensory acceptance, in vivo efficacy and product stability.
Thank you again for your insightful review. Your opinions have greatly enhanced the rigor and scientific nature of our conclusion section.
Round 2
Reviewer 1 Report
Comments and Suggestions for Authors
foods-3897579-peer-review-v2
Authors have improved the manuscript; however, some additional editorial adjustments and corrections needs to be considered. Please, before you submit the revised version, ask some of your senior collaborators to proof read your modified versions. Several technical adjustments needs to be made as consequence of the corresponding author (maybe) negligence.
ln 55 and etc. Word lactobacilli needs to be without capital L and without italics. Italics is applied for not English words (as Latin names, and other words in different from English).
Ln62: Correct to Ls. casei
Ln142: TPC was introduced on Ln135.
For all abbreviations used in the text, when for first time was introduced, needs to be explained. In following occasions, please, use that already introduced abbreviations. Please, check entire manuscript.
Ln172: centrifugation need to be in g force.
Please, check entire manuscript.
Ln184: abbreviations for Gama-aminobutyric acid is GABA.
States names needs to be abbreviated. Please, check and correct entire manuscript.
Ln408: Correct name is Lactiplantibacillus plantarum. Since already this species was mentioned in the text, in this occasions need to be abbreviated.
Author Response
Dear Reviewer,
We sincerely appreciate the time and effort you have dedicated to reviewing our manuscript (Manuscript ID: foods-3897579) and providing us with such valuable and constructive comments. Your insightful suggestions have significantly contributed to enhancing the academic quality of our work.
In response to your feedback, we have carefully revised the manuscript accordingly. All modifications have been highlighted with yellow for your convenience. Below, please find our point-by-point responses to each comment, along with the corresponding revisions in the resubmitted files.
Comment 1
In 55 and etc. Word lactobacili needs to be without capital L and without italics. ltalics is appliedfor not English words (as Latin names, and other words in diferent from English).
Response 1
We sincerely thank the reviewer for this meticulous comment regarding the proper formatting of microbial nomenclature. We have thoroughly reviewed the manuscript and corrected the instances where the common name “lactobacilli” was incorrectly formatted. Specifically, we have ensured that:
The common English name “lactobacilli” (and its singular form “lactobacillus”) is written in lowercase and in Roman (non-italic) font throughout the text.
The italicization is strictly reserved for Latin genus and species names (e.g., Lacticaseibacillus casei, Ls. casei, Lactiplantibacillus plantarum).
These changes have been made in the relevant sections of the revised manuscript.
Comment 2
Ln62: Correct to Ls.casei
Response 2
We sincerely appreciate the reviewers' valuable comments. In the revised manuscript, we have corrected the error regarding Ls.casei and thoroughly reviewed the entire text to ensure accuracy. We deeply regret our oversight and thank the reviewers for bringing this issue to our attention.
Comment 3
Ln142:TPC was introduced on Ln135
Response 3
We sincerely appreciate the reviewers' valuable comments. In the revised manuscript, we have corrected the error.
Comments 4
For all abbreviations used in the text, when for first time was introduced, needs to be explained In following occasions, please, use that already introduced abbreviations. Please, check entiremanuscript.
Response 4
Thank you for your reminder. In the revised manuscript, we have carefully reviewed and standardized all abbreviations throughout the text. Particularly in sections related to "fermented pineapple juice," the full term has been consistently used, and all instances of the previous abbreviated form have been removed.
Comments 5
Ln172:centrifugation need to be in g force
Response 5
Thank you for your reminder.We had changed “rpm”to“g”in the revised manuscript(Ln170).
Comments 6
Please, check entire manuscript.
Response 6
Thank you for reviewing our manuscript and providing your valuable feedback. In accordance with your suggestions, we have conducted a thorough line-by-line review of the entire text and implemented comprehensive revisions to address issues related to grammar, spelling, formatting, and consistency. The specific modifications are summarized as follows:
- Language and Grammar Refinement
We have revised the language throughout the manuscript to improve clarity, accuracy, and fluency in alignment with the conventions of academic English writing. Key changes include:
Correction of minor grammatical errors, particularly in the use of prepositions and articles.
- Formatting and Consistency Improvements
We have ensured uniformity in formatting across the entire manuscript:
Microbial nomenclature: All strain names (e.g., *Ls. casei*, *L. plantarum*) are now consistently italicized in accordance with scientific naming conventions.
Spacing: Standardized spacing between parentheses, numbers, and units. For example, "(Fig. 2A)" has been corrected to "(Fig. 2A)" and "0.5mL/min" to "0.5 mL/min".
- Figures and References
Figure and table citations: Verified the accuracy and consistency of all figure and table references (e.g., Fig. 2, Table 1) in the main text.
Reference formatting: Conducted a comprehensive review of the reference list to ensure compliance with journal guidelines. For instance, the volume number format in Reference 11 has been corrected accordingly.
Comments 7
Ln184:abbreviations for Gama-aminobutyric acid is GABA
Response 7
Thank you for your thorough review and insightful suggestions.We have carefully revised the manuscript in accordance with your recommendations.
On line 183, "Gamma-aminobutyric acid" has been updated to "Gamma-aminobutyric acid (GABA)", and all instances of "g-ABA" in Table 1 and Figure 6 have been corrected to "GABA" for consistency and accuracy.
Comment 8
States names needs to be abbreviated. Please, check and correct entire manuscript
Response 8
All US state names throughout the manuscript, including in the Materials and Methods and References sections, have been consistently abbreviated in compliance with journal guidelines. For instance, "Missouri, USA" has been revised to "MO, USA" and "California, USA" to "CA, USA". All such revisions have been highlighted in yellow in the manuscript.
Comment 9
Ln408: Correct name is Lactiplantibacillus plantarum. Since already this species was mentionedin the text. in this occasions need to be abbreviated.
Response 9
We thank the reviewer for this careful observation. According to the standard microbiological nomenclature, we have now abbreviated Lactiplantibacillus plantarum to Lp. plantarum at line 55 and all other subsequent mentions in the manuscript after its first full introduction.
Reviewer 2 Report
Comments and Suggestions for Authors
The authors have satisfactorily addressed all my comments, resulting in an improved manuscript.
Author Response
We read comments carefully and have made correction which we hope meet with approval. Revised portion are marked in red in the paper. Special thanks to you for your good comments. And I would request you to consider it for publication in your journal. Thank you very much.
Reviewer 3 Report
Comments and Suggestions for Authors
This paper can be accepted for publication.
Author Response

(The authors gave the same response as above.)
